# Reasoning and Logical Proofs of the Fundamental Laws: “*No Hope*” for the Challengers of the Second Law of Thermodynamics

**DOI:** 10.3390/e25071106

**Published:** 2023-07-24

**Authors:** Milivoje Kostic

**Affiliations:** Department of Mechanical Engineering, Northern Illinois University, DeKalb, IL 60115, USA; kostic@niu.edu

**Keywords:** fundamental laws, second law of thermodynamics, Carnot cycle, reversible equivalency, exergy, entropy generation, thermal roughness, thermal friction irreversibility, thermal transformer, virtual thermal particles

## Abstract

This comprehensive treatise is written for the special occasion of the author’s 70th birthday. It presents his lifelong endeavors and reflections with original reasoning and re-interpretations of the most critical and sometimes misleading issues in thermodynamics—since now, we have the advantage to look at the historical developments more comprehensively and objectively than the pioneers. Starting from Carnot (*grand-father* of thermodynamics to become) to Kelvin and Clausius (*fathers* of thermodynamics), and other followers, the most relevant issues are critically examined and put in historical and contemporary perspective. From the original reasoning of generalized “energy forcing and displacement” to the logical proofs of several fundamental laws, to the ubiquity of thermal motion and heat, and the indestructibility of entropy, including the new concept of “thermal roughness” and “inevitability of dissipative irreversibility,” to dissecting “Carnot *true reversible-equivalency*” and the critical concept of “thermal-transformer,” limited by the newly generalized “*Carnot-Clausius heat-work reversible-equivalency* (CCHWRE),” regarding the inter-complementarity of heat and work, and to demonstrating “*No Hope*” for the “*Challengers*” of the Second Law of thermodynamics, among others, are offered. It is hoped that the novel contributions presented here will enlighten better comprehension and resolve some of the fundamental issues, as well as promote collaboration and future progress.

*“If your theory is found to be against the second law of thermodynamics, I give you no hope; there is nothing for it but to collapse in deepest humiliation”.*—Arthur Eddington

Impasse: *“Perhaps, after all, the wise man’s attitude towards thermodynamics should be to have nothing to do with it. To deal with thermodynamics is to look for trouble”.*

Anecdotal Laws of Thermodynamics (LT) *[bracketed terms added]:♦[0LT]: You must play the game [equilibrium]. ♦[1LT]: You can’t win [conservation]. ♦[2LT]: You can’t break even [dissipation]. ♦[3LT]: You can’t quit the game [0 K impossible]*.—Thermodynamics-WikiQuote

*“The Second Law of thermodynamics can be challenged, but not violated—Entropy can be decreased, but not destroyed at any space or time scales. […] The self-forced tendency of displacing nonequilibrium useful-energy towards equilibrium, with its irreversible dissipation to heat, generates entropy, the latter is conserved in ideal, reversible processes, and there is no way to self-create useful-energy from within equilibrium alone, i.e., no way to destroy entropy”.*—[2LT.mkostic.com (accessed on 30 June 2023)]

## 1. Introduction

This treatise aims to present the lifelong endeavors and reflections, including additional, original reasoning and interpretations by this author, regarding the fundamental issues of *Thermodynamics,* especially as related to the subtle *Second Law of Thermodynamics* (2LT) [1]. It is written for the Special Issue of the *Entropy* journal dedicated to this author’s 70th Birthday [2].

Science and technology have evolved over time on many scales and levels, so we now have the advantage to look at its historical developments more comprehensively and objectively than the pioneers [3,4,5]. As Anthony Legget, a Nobel laureate, commented [6], “Mathematical convenience versus physical insight […] that theorists are far too fond of fancy formalisms which are mathematically streamlined but whose connection with physics is at best at several removes […] heartfully agreed with Philippe Nozieres that ‘*only simple qualitative arguments can reveal the fundamental physics*’.” In that regard, mostly physical and simple qualitative insights will be examined and emphasized here.

The goal has been to examine and scrutinize the ambiguous, interwoven, and challenging issues in thermal science, and to present some novel contributions, with the hope to resolve a number of unsettled issues, and to encourage constructive criticism and collaboration for further progress. More specific and elaborate publications by this author and others are anticipated in the future.

In addition to the original interpretations, the following, more specific and novel concepts are offered here: synergy of generalized and conjugate “energy forcing-and-displacement”; logical proofs of several fundamental laws; ubiquity of “thermal-roughness” as a new concept; reasoning infeasibility of entropy destruction and inevitability of irreversible work potential dissipation; conjugation of work dissipation and entropy generation; “thermal-transformer” concept, governed by newly generalized, Carnot–Clausius heat–work reversible equivalency (CCHWRE); the impossibility of the 2LT violation at any space and time scale (for which ThD macro-properties are defined), without exception, among others.

The diverse and perplexing terminology and definitions (in different branches of science) contribute to further ambiguity and confusion, and sometimes misunderstanding. Due to the lack and inadequacy of specific scientific vocabulary, some thermodynamic terminology is emphasized and synergized here by *uncommon connotations*, by using “dashed-attributes” with the respective nouns, “quoting words”, and similar, *in order to emphasize thermodynamic-meaning,* as distinction from the meaning of the common terminology. The selected assertions are emphasized throughout this treatise under “*Key Points*” while questionable (deficient) statements and misrepresentations are underscored as “*Challenge Points*” and “*False Points*,” respectively.

Thermodynamics, as the science of energy and entropy, is the most fundamental discipline, and as such, it encompasses all existence in space and transformations in time, in nature. As stated by Gyftopoulos and Beretta [7], “Thermodynamics is not a tree-branch of physics. It pervades the entire tree. To emphasize this conception, we often use the words physics and thermodynamics as synonyms.” Due to the complexity of the diverse natural and artificial systems and processes, the fundamental laws often appear elusive and sometimes mystified. It is hoped that the logical reasoning presented here will contribute to improved comprehension of the fundamental concepts and related laws of thermodynamics and nature.

The fundamental *Laws of Thermodynamics* (LT) are the fundamental laws of nature, and they are considered to be axiomatic and experiential without proof, as never experienced otherwise, or as self-evident postulates. Due to the very complex micro- and macro-structures and their intricate interactions, it would be impossible to deterministically prove the *Laws*, but they could be reasoned logically, and their general validity inferred in principle, as it will be deduced here.

Since all existence is in principle mechanistic and physical, it will be demonstrated here that the LT are generalized extensions of the fundamental Newton’s Laws (NL) of mechanics. The First Law of Thermodynamics (1LT) is the generalized law of the conservation of energy, and the Second Law of Thermodynamics (2LT) describes the forcing tendency of non-equilibrium, useful energy (or work potential, WP) for its displacement and unavoidable, irreversible dissipation to heat with entropy generation, towards mutual equilibrium.

The content of this treatise is presented in several sections, see Appendix A. After the *Introduction* in Section 1, in the following Section 2, “*Energy forcing and displacement*”, the related concepts are pondered. The “force or forcing” is non-equilibrium energy tendency to displace or redistribute (or to extend) from its higher to lower energy density (or *energy intensity*; see Table 1) towards mutual equilibrium with uniform properties. Then, Section 3, “*Reasoning logical-proof of the fundamental laws*,” reveals the concept of energy displacement as the *mechanistic phenomenon* in general, where the elementary particles (including “field-equivalent” particles) or bulk systems (consisting of elementary particles), mutually interact along shared displacement (with equal, respective action–reaction forces), thereby conserving energy during their interactive, mutual displacement. Then, in Section 4, “*Ubiquity of thermal motion and heat, thermal roughness, and indestructability of entropy,*” this author’s comprehension of related phenomena is further advanced by defining a new concept of “*thermal roughness*” and reasoning impossibility of entropy destruction, among others. In Section 5*,* “*Carnot maximum efficiency, reversible equivalency, and work potential,*” the Sadi Carnot’s ground-breaking contributions of reversible processes and heat-engine cycles’ maximum efficiency is put in historical and contemporary perspective, and it is argued that the Carnot’s contributions are among the *most important developments* in natural sciences. In the succeeding Section 6, named here “*Thermal Transformer: Carnot–Clausius Heat–Work Reversible Equivalency*” concept*,* a notion of *true* “*heat-work inter-complementarity*,” is articulated and named here, as an essential consequence of “*true*” *reversible equivalency*. Lastly, Section 7, “‘*No Hope*’ *for the Challengers of the Second Law of thermodynamics*,” presents this author’s compelling arguments that “entropy can be reduced (locally, when heat is transferred out of a locality), but it cannot be destroyed by any means on any space or time scale of interest.” Entropy, as the “*final transformation*” cannot be converted to anything else nor annihilated, but only transferred with heat and irreversibly generated with heat generation due to work dissipation, including Carnot “thermal work-potential” dissipation. Relevant conclusions are presented in Section 8.


**
*Selected Essential Abbreviations and Notes*
**
*(in logical order for usage convenience):*


ThD: *Thermodynamics* (Carnot’s concept of “maximum power [work] from heat” in 1824, name coined in 1854 by William Thompson, named Lord Kelvin).

ThM: *Thermal motion* (“*vis-viva*”: Lomonosov 1738, Count Rumford [Benjamin Thompson] 1798, Brown 1827, Clausius 1857, Maxwell & Boltzmann 1859). 

ThT: *Thermodynamic (absolute) temperature* (1848 Lord Kelvin [W. Thompson]): based on the Carnot concept, and 1854 via the IG derivation).

ThP; *Thermal particles* are conserved physical particles, such as atoms, molecules, electrons, and similar, that undergo thermal interactions via chaotic thermal motion and collisions.

ThVP; *Thermal virtual particles* [as named here] are non-conserved dynamic particles (as distinguished to physical ThP); they increase with entropy increase, i.e., with an increase in thermal randomness of the physical-ThP.

*N_ThVP_: Number of Thermal Virtual Particles*, ThVP. It may be considered as the “*particle dimensionless entropy.*”

*n_th_*: *Number of thermal moles* is the number of ThVP per the Avogadro’s number, i.e., *n_th_*= *N_ThVP_/N_A_.* It may be considered as the “*molar dimensionless entropy.*”

*Self* or *spontaneous* is a *self-driven process* within the interacting systems, without any external forced influence, i.e., without any “external compensation”.

*Dissipation* is any “frictional” conversion of work or work potential (WP) into heat or thermal energy, with diminished remaining useful WP, resulting in irreversible energy degradation and commensurate entropy generation.

Irr: *Irreversibility* is “irreversible transformation”, or something permanently changed (accompanied by irreversible entropy generation), without possibility to fully (or “truly”) reverse all interacting systems back by any means (impossibility of entropy destruction or annihilation). It should not be confused with local change back to the original condition by “compensation” from elsewhere [8]. 

CIrr: *Complete irreversibility* is the case when all work potential (WP) within all interacting systems is dissipated to heat (without any work extraction) with maximum possible entropy generated after the mutual “steady equilibrium state” is achieved. The final, mutual equilibrium state (without mutual WP) is independent on the quality of the initial energy since all WP would be dissipated, but it may have residual WP with regard to the other surrounding systems not in equilibrium with the mutual equilibrium system.

*Thermal roughness* (and related *Thermal friction*) (as named here) are the underlying cause and source of unavoidable irreversibility (2LT) since absolute-0K temperature is unfeasible (3LT), i.e., a perpetual, real “smooth surface” is impossible due to perpetual and unavoidable, dynamic ThM of ThP.

*Reversibility* or *reversible equivalency* is an *ideal concept*, represented by *ideal processes* without any energy degradation (with maximum possible efficiency or without irreversible dissipation) so that their output and input are *truly equivalent* and may self-reverse-back completing a cycle, or may perpetually repeat back-and-forth in any manner, therefore, effectively representing “*dynamic (quasi-) equilibrium.*” [9].

*Carnot cycle* is an ideal, reversible cycle (with maximum possible, 100% *2LT efficiency*) consisting of reversible heat transfers and isentropic work transfers, extracting maximum WP between the two reservoirs at high and low temperatures. Therefore, as the “work-extraction measuring-device”, its cyclic efficiency is the measure of the WP between the two reservoirs only, i.e., it is dependent on the two temperatures only, and it is not dependent on its design or mode of operation (independent of the quasi-stationary cyclic path or any other, reversible stationary process path), i.e., it is not dependent on the cycle per se.

CtEf: *Carnot Efficiency* (a.k.a. *Carnot cycle theorem or Carnot function*) is the maximum possible, reversible cycle efficiency interacting within the high- and low-temperature reservoirs (*η**_max_** = η**_C_** = W**_C_**/Q**_H_** =* [1 − *Q**_L_**/Q**_H_**]**_rev_** = F**_C_**(T**_H_**,T**_L_**) =* 1 − *T**_L_**/T**_H_***), see “*Carnot Cycle*” above.

CtEq: *Carnot Equality* (as named here, to resound the integral *Clausius equality,* being its precursor) is the heat-temperature ratio equality for reversible cycles between any two thermal reservoirs (*Q**_L_**/Q**_H_*** = *T**_L_***/*T**_H_***, or *Q**_H_***/*T**_H_*** =*Q**_L_**/T**_L_*** = *Q***_ref_***/T***_ref_** =***Q/T***
**=**
*constant*).

CsEq: *Clausius Equality* of cyclic integral for any reversible cycle (Cycle *Integral*[*δQ/T*] = 0 is deduced from the *Carnot Equality, CtEq*).

CCHWRE: *Carnot–Clausius Heat–Work Reversible Equivalency,*” (as emphasized and named here) is a generalized concept of heat–work interchangeability as an essential consequence of *“true” reversible equivalency*.

*Thermal transformer*, combined power-heating and refrigeration cycles (including heat-pump cycles), devices that transfer heat from any to any temperature level, governed by the CCHWRE concept.

WP: *Work potential* or maximum possible *useful energy* of a non-equilibrium system with regard to its reference equilibrium (i.e., Carnot’s motive power of heat, sometimes *work* for short), or related *free energy*, or *exergy* (where the surrounding reference is standardized).

IG: Ideal gas (*PV/(t + C) = k* Clapeyron in 1834; *PV = nRT* Renault in 1845). 

1NL: *First Newton Law* of equilibrium motion or resting inertia.

2NL: *Second Newton Law* of forced change of momentum or acceleration.

3NL: *Third Newton Law* of action and equal reaction (duality of balanced forces and conservation of momentum).

0LT: Zeroth Law of Thermodynamics (temperature uniqueness of thermal equilibrium).

1LT: First Law of Thermodynamics or First Law for short (energy conservation; 1843 Joule, 1847 Helmholtz).

2LT: *Second Law of Thermodynamics* or *Second Law* for short (*non-equilibrium useful-energy dissipation with entropy generation towards equilibrium*; 1824 Carnot, 1850 Clausius, 1851 Thompson, 1854 Clausius theorem (dQ/T), 1865 Clausius entropy, 1874 Clausius formal statement of 2LT, 1867 Maxwell’s Demon, 1876 Gibbs free energy in Chemical ThD). 

3LT: *Third Law of Thermodynamics* (*impossibility of* (thermal) *emptiness* (impossible absolute-0K nor to stop thermal motion); 1906 Nernst).

4LT: *Forth Law of Thermodynamics* (*impossibility of evolution forever,* “growth without decay” is impossible; selective and self-reproductive evolution is extending inevitable irreversibility. Note that the 4LT is evolving in many forms and it is still to be defined!). 

PMM0 or PM: *Perpetual-motion machine of the zeroth kind* (or “*Perpetual [free] motion*” in short) that violates the irreversible dissipation or friction (impossibility of free perpetual motion without dissipative resistance).

PMM1: *Perpetual-motion machine of the first kind* that violates the 1LT (impossibility of creating energy from nowhere).

PMM2: *Perpetual-motion machine of the second kind* that violates the 2LT (impossibility of self-creating useful-energy or WP from within equilibrium).

PMM3: *Perpetual-motion machine of the third kind* that violates the 3LT (impossibility of converting all heat to work since absolute-0K temperature is unachievable).

PMM4: *Perpetual-motion machine of the fourth kind* that violates the 4LT (impossibility of evolution forever without decay, or similar: note that the 4LT is evolving in many forms and it is still to-be-defined!).

## 2. Ubiquity and Conjugation of “Energy Forcing and Displacement”

The *mass energy* of material systems (or *energy* for short) is non-uniformly distributed (or displaced) within the systems’ energy space (or displacement space, or *displacement* or *extensity* for short), with non-uniform *energy density*, i.e., energy per unit of its displacement space (or energy *intensity* or energy force, or *force* for short). The *displacement* is the energy extensive property, and by definition, it is the conjugate with its energy *force*; see Table 1.

There is a natural, directional *forced tendency* to displace energy from higher to lower *intensity* and during such a process (displacing energy from higher to lower intensity locality), it equalizes the energy intensity (or energy density) while asymptotically approaching the stable mutual equilibrium of balanced forcing and infinitesimal fluxes within all interacting systems.

Namely, an acting particle (or body, a bulk of particles in general) with higher energy density interacting with a body at lower energy density, the two being in non-equilibrium (non-equal energy densities), will act to displace its energy (acting body’s energy) onto a resisting (reacting) body, resulting in decreasing acting-body energy and energy density (“figuratively decelerating”) while increasing the reacting body’s energy and energy density (“figuratively accelerating”) until the energy density of the interacting bodies equalize and forcing-interactions cease, when mutual state of stable equilibrium is reached, without further energy displacement.

This natural phenomenon is the meaning and origin of *forced tendency* (definition of forcing or *force*), as well as the meaning of the *energy displacement*, i.e., description and formulation of the *laws* of mechanics and thermodynamics. There is a deep meaning behind the vocabulary and description of the fundamental concepts to be elaborated elsewhere.

**Table 1 entropy-25-01106-t001:** Typical Energy *Intensive* and *Extensive* Conjugate Properties (*Energy Force* and *Energy Displacement*).

*Generic Name*	*Customary Name*	*Definition*	*Unit*
**Energy Force** (or ***intensity***) (*intensive* property, conjugate with energy displacement)	Generalized force (*intensity*)	Energy intensity or energy density is energy per unit of energy displacement, by definition, it is the conjugate property with energy displacement, see next.	[*F*] =J/[***δ***]
**Energy displacement** (or ***energy space*** or *extensity*) (*extensive* property conjugate with energy force)	Generalized displacement (*extensity*)	Energy extensity or energy space is energy per unit of energy intensity, by definition, it is the conjugate property with energy force, see above.	[***δ***] see specifics below
Mechanical force (Newtonian)	Force (Newtonian)	Newtonian bulk force or total pressure force, or energy per unit of bulk displacement.	N = J/m
Mechanical displacement (Newtonian)	Displacement (linear)	Linear displacement of bulk body or Energy per Newtonian bulk force.	m
Mechanical compression force	Pressure	Mechanical compression energy per space volume.	J/m^3^ =N/m^2^
Mechanical compression displacement	Volume	Compressible volume.	m^3^
Thermal force	Temperature	Thermal energy per unit of entropy (or average thermal energy per dynamic thermal particle).	K (or J/[k_B_] = J/[1] ^(+)^)
Thermal displacement ^(^*^)^	Entropy or number of thermal virtual particles	Thermal energy per absolute temperature (or number of dynamic, *thermal virtual-particles*; irreversibly generated, include thermal-particle chaotic-dynamics in space, non-conserved).	J/K (or [1])
Chemical force	Chemical potential	Chemical energy per unit of number or moles of species (or per number of chemical species).	J/Mole =J/[1]
Chemical displacement	Number of moles or species	Number of species or number of moles of chemical species (conserved).	[1]
Electrical force	Voltage	Electrical energy per unit of electrical charge (or per number of charged particles).	V = J/C (or J/[1])
Electrical displacement	Capacity or number of charged particles	Electrical energy per unit of electrical force (or number of electrically charged particles; conserved).	C = J/V (or [1])
Etc., for other energy types (the above are not inclusive)	Etc.	Magnetization, nuclear, radiation, etc.	Etc.

^(+)^ NOTE that [1] is dimensionless unit and not a Reference. ^(^*^)^ NOTE that all but thermal energy displacements are conserved, while thermal displacement (entropy or number of thermal virtual particles, *N_ThVP_*) is irreversibly generated due to dissipation of all other energy types to heat.

***Key Point 1*.** *Mass energy,* or *energy* in general, is the underlying, building block of all energy fields and material existence in space (“activity” of all fundamental particles, including field-equivalent particles, and “inertia” of their bulk interactions) with a spontaneous tendency to displace in time towards mutual, stable equilibrium, thus defining space and time existence. During its displacement, energy is conserved (1LT).

***Key Point 2.*** *Force or forcing is the spontaneous (by-itself or of-itself) energy tendency to displace*, directionally from higher to the “adjoining” locality of lower intensity (from higher to lower energy density). Since displacement is the mutual interaction of competing particles and systems, the force duality is mutually exhibited and balanced between the interacting systems, the action and reaction forces as described by the *Third Newton Law* (3NL), including the acceleration force (the *Second Newton Law*, 2NL), and including its special case of uniform motion without acceleration (uniform velocity, including zero velocity or resting), with balanced external forces as described by the *First Newton Law* (1NL).

***Key Point 3.*** *Useful-energy* or *work-energy potential* (or free energy, or work potential, or *work* for short) is the non-equilibrium energy within interacting systems, capable of displacing spontaneously (by itself) out of a system while it is coming at the mutual equilibrium with the most efficient processes (without dissipative conversion to heat). In an ideal reverse-process, such original work, as the *formation work*, would create the original non-equilibrium. If the surrounding reference system is well defined (*P**_0_**, T**_0_**, µ**_0,i_***, *V**_0_**, …*), then such *work potential* (WP) of a given system state (*P, T, µ**_i_**, V,* …) is a unique [quasi-] property of the system state and is defined as *exergy*. Therefore, *useful energy*, *work potential,* or *exergy* are essentially the same concepts and conserved during ideal, reversible interactions. In real, irreversible processes, the work (i.e., *exergy*) will be dissipated (converted) to heat and diminished. The WP as energy cannot be generated but only displaced (transferred) and is not conserved since it is irreversibly dissipated to heat with entropy generation (2LT).

***Key Point 4.*** The *driving cause and source* of any and all process forcing, manifested by energy displacement, is due to non-equilibrium WP, or the exergy difference between any two process states.

***Key Point 5.*** The *energy process* (i.e., energy interaction displacement, or *process* for short) is caused or driven by *directional forcing* due to non-equilibrium WP. Ideally, in the most possible efficient, reversible processes, the WP (or exergy) is conserved; however, in real processes, the exergy is dissipated to heat with *entropy generation* due to diverse causes of directional work dissipation, i.e., chaotic energy redistribution in all possible directions, known as dissipation of WP into randomized thermal-energy, or dissipation of work to heat during a process. If all WP is dissipated, then the mutual equilibrium is achieved with no mutual work potential, with maximum entropy, and with no possibility of any further energy displacement, unless external exergy (i.e., WP) is applied.

***Key Point 6.*** There is no perfect equilibrium, nor perfect absolute zero temperature, nor reversible process, nor any other ideal, perfect system nor process. However, such perfect systems and ideal processes are very useful and often necessary to describe and define fundamental concepts of natural phenomena, and to quantify properties and relevant equivalences.

## 3. Reasoning Logical-Proofs of the Fundamental Laws

The two fundamental *Laws of Thermodynamics* (1LT and 2LT) are believed to be empirical and axiomatic without proof. However, they are *mechanistic in nature* and in principle are more general consequences of the Newtons’ law of motions, see Figure 1. The three *Newton’s Laws* (NL) of forces and motions are holistic in a sense that the 2NL of motion is also the 3NL of action–reaction equality when the inertial forces are included, and the 1NL of inertia is a special case of the 2NL when external forces are balanced (zero).

It may be deduced that, due to equality of acting and reacting forces along the same mutual displacement, the energy transferred by acting-force displacement must be equal to the energy of the opposing, reacting-force displacement—that is, the interactively displaced (or transferred) energy from acting to reacting particle or body will be conserved. In the absence of the opposing reaction forces, there will be no energy transfer. This will be true in general since all elementary and/or bulk interactions are additive, regardless of complexity of system structure or types of interactions (also forced fields could be represented by relevant “equivalent particles,” such as photons, etc.).

Furthermore, it is reasoned here that the energy directional transfer (2LT) is due to a particle or body forcing action onto another particle or body resisting to change its existential “*inertial-state”*, by equal reacting force in opposite direction (the 3NL) along a mutual displacement.

As shown on Figure 1, since the *action force*, *F**_A_***, is balanced by (equal magnitude to) *reaction force*, *F**_R_***, (the 3NL, including inertial force—the 2NL) along the shared, *interaction displacement*, *dL**_A_*** = *dL**_R_***, then, the amount of “action energy out” would equal to the amount of “reaction energy into”, i.e., the energy is conserved during any and all interactions (First Law of Thermodynamics, 1LT):(1)$$\left\{\left|dE_{A}=F_{A}dL_{A}\right|\right\}=\left\{\left|F_{R}dL_{R}=dE_{R}\right|\right\}$$

For “free motion” or “free expansion” no energy transfer (energy is conserved within), i.e.,
(2)$${(F}_{R|A}=0,\;\;dF_{R|A}=0)\;\mathrm{and}\;dE_{R|A}=0\cdot dL_{R|A}=0$$

Therefore, during the mutual (shared and equal) displacement, the acting body will be transferring its energy onto the reacting body, the two being the same, in principle, the product of equal action and reaction force (including process inertial forces) and equal mutual displacement. Therefore, the directional energy transfer and dissipation (2LT) and energy conservation (1LT) are consequences of the fundamental Newton’s Laws of mechanics, and not merely empirical as commonly postulated, see Figure 1 and Equations (1) and (2).

***Key Point 7.*** All interactions in nature are mechanistic, and during forced interactions, energy is directionally transferred (2LT) and conserved (1LT). In cases without interaction, if a particle (or a body, a bulk system) is bounded by an enclosure boundary (thus restricting displacement), or not encountering resisting particle (or resisting body; no reaction force), the particle or body will stay at rest or continue with its “free motion,” or an expanding gas without any resisting interaction will undergo “free expansion” without transferring any energy, and therefore, the energy will be conserved internally within (Figure 1).

***Key Point 8*.** The forced-displacement interaction is a process of energy transfer from the acting particle (or body) with higher energy density onto a reacting particle (or body) of lower energy density, displacing (transferring) its energy during the interaction, i.e., diminishing its energy (figuratively “decelerating” its structure) while increasing energy of the reacting body (figuratively “accelerating” its structure) until the energy densities (or intensities) are equalized when mutual self-sustained equilibrium is achieved.

In addition to reasoning the physical concepts of the 1LT and 2LT laws, further inferences and/or reasoning proofs for new or newly re-interpreted concepts are also offered throughout, i.e.,: “Thermal Virtual-particles” and “Thermal-moles” as dimensionless entropy (Section 4.2); Heat–work energy “Pond analogy” misconception (Section 4.3); Ubiquity of “Thermal-roughness & thermal-friction” (Section 4.4); Inevitability and Conjugation of Work-dissipation and Entropy-generation (proving the Planck’s statement to be misplaced, Section 4.5); Carnot Equality (Section 5.2); Thermal transformer and *Carnot–Clausius Heat–Work Reversible Equivalency* (CCHWRE, Section 6.1); Proof of Ideal gas state (Section 6.2); Reversible Cycle efficiency is perfect (100%) and essentially “measure” the WP of heat (Section 6.3); Primary “2LT Deception structures” (Section 7.2); “Thermodynamic paradox” demystified (Section 7.3); among others.

## 4. Ubiquity of Thermal Motion and Heat“, Thermal-Roughness,” and Indestructibility of Entropy

### 4.1. Ubiquity of Thermal Motion, Thermal Energy and Heat, Temperature, and Entropy

*Thermal motion* (ThM), *thermal energy,* and *heat* are ubiquitous and are perpetually generated by *work potential* (WP) transfer and storage (hence during all processes) where the WP is in part irreversibly dissipated to heat (in principle, increasing ThM and thermal energy, i.e., temperature and/or entropy). The WP dissipation is caused by different types of “dissipative-frictions,” ultimately instigated by “*thermal roughness*” (as reasoned, defined and named here) due to the existing, chaotic ThM of thermal particles (ThP). Furthermore, since the ThM cannot be ceased (i.e., zero absolute temperature is unattainable; the 3LT), the dissipative irreversibility is unavoidable in general (the 2LT), contributing to further ubiquity of heat and related thermal phenomena, for all processes without exception.

***Key Point 9.*** As an adjective, “*thermal*,” implies a chaotic, randomized motion, kind of “thermal turbulence.” Average thermal energy per particle is *temperature* (or *intensity* of ThM energy), and extensive randomness of the bulk ThM is *entropy* (or *extensity* of ThM energy; or the total *ThM energy per temperature*, since intensity and extensity are the conjugate thermal-energy properties, see Table 1).

*Thermal energy* (or *stored-heat* or energy of ThM) is transferred (or displaced) as *heat* via ThM and thermal collisions, from the ThP at a higher temperature on average, to the ThP at a lower temperature on average, by means of random “poking or jiggling” of ThP, across a real or imaginary boundary, without the need for physical ThP to be displaced across, similarly to how the AC electrical energy is displaced (but in an orderly, wavy or cyclic manner, not as with chaotic thermal energy) from one electron to another, without need for electron displacement *per se*.

*Temperature* is thermal intensity or thermal force, i.e., particle-average thermal-motion energy per thermal particle, such as an atom, molecule, electron, or similar. The temperature is measured by thermometers with calibrated empirical scales (with Fahrenheit or Celsius degrees), including ideal gas thermometers, the latter being used to infer the absolute temperature, with absolute zero being the lowest temperature possible, in-principle, when ThM is seized (absolute zero Kelvin scale is 0 K = −273.15 °C with the same degrees as the Celsius’). The most fundamental, thermodynamic temperature concept, independent of thermometer design, was inferred by Kelvin in 1848 based on the Carnot cycle efficiency, using the same ideal gas absolute temperature scale. The origins and thermodynamic concept of temperature were reviewed by Cropper [10] and elsewhere.

*Entropy* is elusive and sometimes puzzling with temperature since both tend to increase with heat generation and storage, and with heat transfer. However, more entropy means more “thermal space,” represented by non-conserved, “thermal virtual-particles, ThVP” (as defined and named in Section 4.2), even though the physical thermal particles, ThP, are conserved. The increase in ThM energy and its extensive randomness (entropy) is, in principle, complemented with a higher average of ThM energy per physical thermal particle (the higher temperature).

The ThM may be ideally intensified (i.e., thermal energy and temperature increased) by reversible work over conserved thermal space (or conserved entropy), and in reverse when work is extracted (thermal energy and temperature are decreased), while thermal space (entropy) is also ideally conserved—see the “Thermal transformer” and CCHWRE concepts in Section 6 and elsewhere).

### 4.2. Thermal Particles, “Thermal Virtual-Particles,” and “Thermal-Moles” or Dimensionless Entropy

*Entropy* is thermal displacement space, defined as the ThM energy per unit of its intensity (temperature), i.e., it is “thermal-space randomness” of chaotically-moving thermal particles, that is, the “randomly traversed-space” by thermal particles, (*randomness* of both, *space directions and dynamic motions*), due to thermal collisions, and it may be represented by *non-conserved* “*thermal virtual-particles*, ThVP” as defined next.

***Key Point 10*.** *Thermal virtual particles* (ThVP) are non-conserved dynamic particles (as opposed to the conserved, physical ThP) and they increase with entropy increase, i.e., with an increase in thermal randomness of the physical ThP. The Avogadro’s number (*N_A_*) of the ThVP represents a “*thermal mole,*” i.e., both are “*dimensionless entropy*,” per ThVP or the mole, respectively.

We now define the number of *“thermal virtual-particles*” *N_ThVP_*, which may be considered as the “*particle dimensionless entropy*,” i.e.,:*N_ThVP_ = S/k_B_ = ln(Ω) = U_th_/(k_B_·T).*(3)

Then, we may define the number of “*thermal moles” n_th_*, which may be considered as the “*molar dimensionless entropy*,” i.e.,:*n_th_ = N_ThVP_/N_A_ = S/(k_B_N_A_) = S/R_u_*(4)
where, *S* is entropy; *U_th_* is the internal thermal energy; *Ω* is the number of the “possible thermal, microscopic states”; *N_A_* is the Avogadro’s number; *k_B_* is the particle Boltzmann constant; and *R_u_* is the molar, universal gas constant.

The number of *thermal virtual particles, N_ThVP_*, is non-conserved, as opposed to conserved number of physical *thermal particles, N_ThP_* (atoms, molecules, electrons, and similar). The former increases with entropy, i.e., with increase in thermal randomness.

### 4.3. Thermal Energy Is a Distinguished Part of Internal Energy (“Pond Analogy” Demystified)

Heat *Q* and work *W* are considered as “energies in-transfer,” as process quantities and not as properties, since after being stored within a system they appear to “lose identities” and increase the system’s “internal energy (*U* or *E*)”, as if they are not distinguishable after being stored. The latter is argued by some and demonstrated with “Pond analogy” by Callen [11] (p. 20): “Heat, like work, is only a form of energy transfer. Once energy is transferred to a system, either as heat or as work, it is indistinguishable from energy that might have been transferred differently. Thus, although đQ and đW_M_ add together to give dU, the energy U of a state cannot be considered as the sum of “work” and “heat” components … the sum is the energy difference ΔU, which alone is independent of the process.” Cullen continued [11]: “The concepts of heat, work, and energy may possibly be clarified in terms of a simple analogy. A certain farmer owns a pond, fed by one stream and drained by another. The pond also receives water from an occasional rainfall and loses it by evaporation, which we shall consider as negative rain.… In this analogy the pond is our system, the water within it is the internal energy, water transferred by the streams is work, and water transferred as rain is heat. … The strict analogy of each of these procedures with its thermodynamic counterpart is evident.”

Furthermore, in an excellent textbook by Gyftopoulos and Beretta [7] (Ch.5), the concept of “Adiabatic Availability (as property)” and “Available Energy (Ch.6),” are presented and assessed. The former, when maximum possible work is reversibly extracted while a system is coming to a stable equilibrium, adiabatically at constant mass and volume (no mass-energy exchange with the surroundings except for the work extraction via reversible “weight processes,” is presented, while the “Available Energy”, represents the maximum work extracted when the system is coming to equilibrium with a reference reservoir, corresponding to the “Exergy” concept. It implies, as argued here, that the “Internal energy” requires further clarification since the same quantity of an internal energy may consist of different forms and quality of energies, including reversible equivalences of stored heat and work within; see the following:

***False Point 1*:** *Callen’s “Pond analogy”* in which the change in internal energy is independent, whether heat or work is added into a system [11] (p. 20), is misleading and generally erroneous, since the pond water is at the same surrounding T and P, which is not the case if we reversibly store internal energy by heating or working. Adding the same reversible amount of work or heat will result in different forms and quality of energy with different final states (with different entropy, volume, etc.), i.e., different WP to be extracted. Therefore, the quality of internal energy is not “the same form and not independent of the process,” as claimed [11]. Namely, the water streams representing work in “pond analogy” undergo full dissipation (called here “*complete irreversibility*,” such as during the famous “Joule’s 1843 experiments (work-heat equivalency; 1LT only)” or isochoric heating only. Only for the “completely irreversible” processes, the outcome is the same internal energy, regardless of whether either the work or heat source of different temperatures are used, since all WP would be completely dissipated within such a system—however, the claim is erroneous in general.

Therefore, for the same total amount of work and heat added to a system, the work potential (WP) differs depending on the “work” and “heat” amounts (reversibly) stored. The internal energy difference (Δ*U*) is independent quantitatively, but dependent qualitatively on the ratio of work added, reflected with different state properties (V, S, etc.), as detailed elsewhere. This author expressed disagreements with such and similar claims [12,13]. For example, if heat *Q′* is stored at constant volume to a system at initial state (*U_i_, V_i_*, *S_i_*, …), the final state will be (*U′ = U_i_ + Q′*, *V′* = *V_i_*, *S′*≠ *S_i_*, …); however, if the same amount of work *W″* = *Q′* is reversibly stored instead, the final state would be different (*U″* = *U_i_ + W″ = U′, V″≠ V′, S″* = *S_i_*, …). If the processes are reversed back to the original initial state, it would be ideally possible to retrieve the original work *W″* from *U″* but that work could not be obtained from *U′* (even though *U′ = U″*), which proves that the internal energies *U′* and *U″* are not the same quality (not the same states, different WPs and *exergies*; the 2LT), regardless of being the same quantity (same *U*-amounts; the 1LT).

***Key Point 11.*** Claiming that *storing Q′ or W″ (if Q′ = W″) would indistinguishably increase the internal energy U*, is only convenient for easy bookkeeping (it sidesteps difficulties of distinguishing energy quality), but *it is deceptive* since *U′* = *U* + *Q′* and *U″* = *U* + *W″* are *not truly (reversibly) equivalent* (not the same *free energies* nor WPs, see Section 5). Namely, there is the *specific and distinguishable quantitative measures* of stored work, i.e., the *work potential* (*WP or available-energy or exergy, or “stored-work”*) within internal energy (*U*), and of stored heat (*thermal energy*, *U_th_*, or “*stored-heat*”) associated with temperature and entropy (*T*, *S*). Both the *exergy* and *U_th_* (being uniquely defined for a specified reference state) may be considered as (quasi-) properties, to be further elaborated in a separate writing.

Additional difficulties of distinguishing internal energy types are due to coupled dualities (conjugate multitudes) of internal energy types (see Section 6). If work or/and heat are stored, even when *U* is quantitatively the same, the other properties that characterize its quality and true equivalency (such as *V*, *S*, etc., in more complex systems), are not the same. Furthermore, (reversible) heat and work are interrelated, and in that regard “interchangeable” as demonstrated by the Carnot cycle and the *Carnot–Clausius Heat–Work Equivalency (CCHWE)*, as defined and named here (see Section 6.1). For example, the ideal gas energy of thermal motion, *Nk_B_T*, manifests also as mechanical compression energy *PV*, as expressed by its equation of state: *PV* ≡ *Nk_B_T*, see Section 6.2.

### 4.4. Thermal Roughness Ubiquity, and Inevitability of Thermal Friction

The “*Thermal Roughness*” and “*Thermal Friction*” are defined and named here as new concepts, as the underlying cause and source of inevitable irreversibility since absolute-0K temperature is unfeasible (3LT). No ideal systems nor frictionless processes are possible due to the unavoidable thermal motion (ThM) of thermal particles (ThP), including the thermal radiation. Even “superconductivity” at lower temperature must be at least infinitesimally irreversible since any energy flow must interact and be affected by the chaotic ThM collisions of the ThP, regardless of the extent (even if infinitesimally small). It is impossible to have a perpetual “smooth boundary surface” due to ubiquitous and unavoidable, dynamic, and chaotic ThM of the ThP; see Figure 2.

It is remarkable that all existing useful energy, quantified by the WP, the cause, and source of process forcing and energy displacement, dissipatedly convert to thermal motion (i.e., generated heat and entropy). In turn, the latter is the cause and source of the “*thermal-roughness*” and “*thermal-friction*.” Furthermore, all “other types of dissipations” are ultimately caused by the underlying “thermal friction”, by dynamic “thermal roughness” due to chaotic ThM (random fluctuations of ThP).

### 4.5. Inevitability and Conjugation of Work dissipation and Entropy generation

Plank challenged the universality of “dissipation of energy” as related to the entropy generation concept [14] (pp. 103–104). Namely, he stated: “The real meaning of the second law has frequently been looked for in a ‘dissipation of energy’. This view, proceeding as it does, from the irreversible phenomena of conduction and radiation of heat, presents only one side of the question. There are irreversible processes in which the final and initial states show exactly the same form of energy, e.g., the diffusion of two perfect gases (§ 238), or further dilution of a dilute solution. Such processes are accompanied by no perceptible transference of heat, nor by external work, nor by any noticeable transformation of energy. They occur only for the reason that they lead to an appreciable increase in entropy. The amount of “lost work” yields a no more definite general measure of irreversibility than does that of “dissipated energy.” Uffink [15], a historian of science, in his analysis of the 2LT literature, commented on the issue: “Before Planck’s work there were also alternative views. We have seen that Kelvin attributed irreversibility to processes involving special forms of energy conversion. This view on irreversibility, which focuses on the ‘dissipation’ or ‘degradation’ of energy instead of an ‘increase in entropy’ was still in use … Planck’s work extinguished these views”.

With all due respect, this author disagrees with Plank that it is “*The same form of energy* … *nor by any noticeable transformation of energy*” It is only the same quantitative amount of energy but not the same quality, not the same work potential, WP (not the same exergy, etc.) since the entropy *S* is increased due to irreversible entropy generation. It will be proven here that any entropy generation is due to irreversible work dissipation to heat, with respective transformation of energy, without exception. The work or WP may be of any kind (mechanical, electrical, chemical, etc.)

The concept is similar to the ideal gas (IG) free expansion (at constant temperature and constant internal energy), where entropy is increased within, due to irreversible process, without work extraction (all WP dissipated within, defined here as “*Complete-Irreversibility*”) and without external heat transfer. However, during such or similar irreversible processes, the entropy is generated within due to internal energy irreversible transformation with degradation (dissipation of its WP to heat), accompanied with unavoidable entropy generation, while the totality of quantitative energy is conserved. Otherwise, during the reversible expansion, work would be extracted and energy and temperature would decrease.

The “Plank’s diffusion mixing and entropy increase” is further similar to the “thermal mixing” of hot and cold system parts in an adiabatic rigid container (such as “melting of cold ice in warm water” in a thermos with no external work nor heat transfer), where the total energy is invariant (conserved) but with decreased WP (decreased energy quality) while energy quantity is conserved.

***Key Point 12:*** In summary, any irreversible “entropy generation” is caused by and related to “heat generation” due to irreversible work (or WP) dissipation, and vice versa; any irreversible work dissipation to heat is always accompanied with irreversible “entropy generation” at any space and/or time scale, without exception. If entropy is generated during any process, then, to reverse the final to the initial state, the irreversible generated entropy has to be removed from the system, which would require removal of the commensurate heat (thus reduction of internal energy); the latter has to be compensated with external work (ideally equal to the prior work dissipated or even more due to unavoidable process irreversibilities) to make up for the prior work dissipation loss reflected in reduced internal energy after the generated entropy (and commensurate heat) is removed from the system. The heat and entropy generation should not be confused with reversible entropy transfer, like during phase change and chemical reaction in equilibrium processes where entropy is conserved.

***False Point 2.*** Plank’s statements regarding “the same form of energy […] diffusion mixing with appreciable increase in the entropy accompanied by no perceptible transference of heat, nor by external work, nor by any noticeable transformation of energy, [14] (pp. 103–104)” are misleading and erroneous, as well as Uffink’s endorsement of the Plank’s claim: “This view on irreversibility, which focuses on the ‘dissipation’ or ‘degradation’ of energy instead of an increase in entropy was still in use … Planck’s work extinguished these views … [15]). However, “to reverse irreversible diffusion,” external work would be required (regardless of the amount), without exception, to compensate for the WP dissipation loss during the prior diffusion.

***Key Point 13:*** The “*Principle of the increase of entropy*” is complementary with the “Principle of (unavoidable) energy degradation” due to the dissipation of WP to heat accompanied with entropy generation (irreversible “entropy increase, not to be confused with reversible entropy transfer”): *δI_rr_ = δ(WP)_diss_ = δQ_gen_ = TδS_gen_* (for a variable process temperature, the differential quantity, *TδS_gen_*, has to be properly integrated along the process path).

If all WP is dissipated after the systems achieve mutual stable equilibrium (without WP), such process is termed here as “*Complete Irreversibility*”.

### 4.6. Irreversibility of Entropy Generation and Indestructibility of Entropy—Essence of the 2LT

All transformations or processes are caused by the non-equilibrium work potential (WP) and are accompanied by energy forced displacement, either as work (in an orderly way) or as heat (via chaotic ThM and thermal collisions, including the “Carnot WP of heat,” see Section 5).

Ideally, in limit, the heat and work could be displaced or transferred in reversible processes without any dissipative loss of the WP, such as in ideal Carnot cyclic processes. Heat is transferred at infinitesimal temperature difference so that entropy transfer from a thermal source is the same as into a thermal sink (either the thermal reservoirs or the system); therefore, the entropy is conserved. Similarly, the adiabatic, reversible work transfers are ideal processes without any dissipation of WP to heat, i.e., they are not associated with entropy and therefore isentropic.

However, all real processes are caused by the displacement of WP and accompanied by dissipation of WP to heat. If no WP to displace, there would be no process-forcing (no process would be possible) as in a self-sustained, perpetual equilibrium. The dissipation of WP is not an “annihilation loss” *per se*, but its conversion and degradation of work to heat (often figuratively named as “work-loss” or degradation of useful energy), accompanied with entropy generation, the latter commensurate with WP dissipation per relevant absolute temperature.

*Irreversibility*, *Irr*, is the “irreversible loss” of the WP, or more accurately the work dissipative conversion to generated heat, *Irr = W_LOSS_ = W_diss_= Q_gen_*. The work dissipation is directly related to the entropy generation, *S_gen_*, at relevant reference, absolute temperature, *T_ref_*, (the Gouy–Stodola correlation, Equation (5). The work dissipation and related entropy generation are two sides of the same coin (“half empty vs. half full”), i.e.,
{*Irr* = [*W_LOSS_* ≡ *W_diss_*] = *Q_gen_*} = *T_ref_* ·*S_gen_* ≥ 0(5)

***Key Point 14.*** The generated entropy is the irreversible “final transformation”: the “lost or dissipated” work is actually compensated with or converted into the generated heat (the 1LT). Furthermore, along the generated heat, the accompanying generated entropy, conjugate to relevant temperature, is the “final and indestructible quantity” since there is no way (no process possible) to convert entropy into nor to compensate entropy with anything at all, nor to annihilate it—the entropy is truly indestructible, the “final transformation” (the 2LT).

***Key Point 15.*** Since *all real, irreversible processes generate heat and entropy* due to the unavoidable dissipation of work to heat (ultimately instigated by the “*thermal roughness*” as elaborated and named here, Section 4.4), and *all ideal, reversible processes conserve entropy*, then, there are *no other processes left* to miraculously generate WP without a due WP-source forcing and transfer, nor any “*imaginary process*” could destroy (or annihilate) entropy, since *it would be a “self-reversal of dissipation” and contradiction impossibility against the natural forcing—*it would imply *self-generation of non-equilibrium* (*and its WP*); therefore, rendering a *logical proof of indestructibility of entropy* (the 2LT). Therefore, there is no process possible (no heat nor work transfer process) to destroy entropy—the thermal *entropy cannot be converted to anything else* nor destroyed, but it will be always irreversibly generated, without exception, at any relevant space or time scale, where the macro-properties could be defined.

***Key Point 16.*** A non-equilibrium (i.e., its WP) may be increased only by forcing on the expense of another WP, as a necessary WP-source. During such forced interactions the WP in ideal reversible processes would be reorganized, i.e., transferred and conserved (1LT and 2LT), or, in part, it would irreversibly dissipate to heat, i.e., the WP would be irreversibly diminished (2LT)—however, the totality of energy (WP and the generated-heat) would be conserved (1LT, again). Therefore, there is no way to self-create non-equilibrium work potential against the natural forcing towards equilibrium. The former would be a contradiction of the latter.

## 5. Carnot Maximum Efficiency, Reversible Equivalency, and Work Potential

### 5.1. Carnot Cycle Maximum Efficiency: Proof by Contradiction Impossibility

It is the intention here to put, in historical and contemporary perspective, the Sadi Carnot’s revolutionary discovery of “reversible processes and maximum-possible efficiency of heat-engine cycles,” and to show that the Carnot’s contributions are among the most important developments in natural sciences; see *Key Point 17* and Figure 3. Only, based on his ingenious and far-reaching reasoning, that the reversible processes and cycles are equally and the most efficient, it was later possible for Clausius and Kelvin and other Carnot followers to discover critical concepts and laws, and to establish *thermodynamics* (ThD) as a new discipline of natural sciences.

***Key Point 17.*** If the critical and ingenious discoveries by Clausius and Kelvin make them “*fathers of thermodynamic*,” then, Sadi Carnot was the “*grand-father of thermodynamics-to-become*”.

The invaluable concepts of “thermodynamic *reversible equivalency*” and concept of useful “*work potential*” were formalized later by others; however, all are based on the original discovery of Sadi Carnot. Long before the inception of *thermodynamics*, even before the (First) Law of energy conservation was established, Sadi Carnot, in 1824, affirmed the following [3]:

“*The motive power of heat is independent of the agents employed to realize it; its quantity is fired solely by the temperatures of the bodies between which is effected, finally, the transfer of the caloric*”, i.e.,
(6)$${W_{netOUT}=W}_{C[Carnot]}=Q_{H}\cdot f_{C}\left(t_{H},t_{L}\right);\;i.e.,\;{\;\;\eta }_{C}={\left.\frac{W_{netOUT}}{Q_{H}}\right|}_{Max|Rev.}=\underset{\mathrm{Qualitative}\;\mathrm{function}}{\underbrace{f_{C}(t_{H},t_{L})}}$$
where, *η_C_* is the Carnot cycle efficiency, maximum possible and equal for any and all reversible cycles. The cycle is converting heat, *Q_H_*, from high-temperature thermal reservoir at *T_H_*, extracting cycle work, *W_C_*, and passing heat to a low-temperature thermal reservoir at *T_L_*.

***False Point 3.*** Some references cite that Sadi Carnot derived the maximum cycle efficiency, *η_C_* = 1 − *T_L_/T_H_*, named in his honor, is false since Carnot wrongly assumed conservation of caloric (*Q_L_* = *Q_H_*), and the absolute temperature were not defined in his time. Regardless, Carnot ingeniously, considering the knowledge at his time, deduced completely and correctly, although implicitly, that the efficiency depends on the two thermal reservoirs’ temperatures, *t_H_* and *t_L_*, only; see Equation (6). The explicit, maximum cycle efficiency was derived later by Kelvin [4] (1850, using IG) and generalized by Clausius [5] (1854), based on Carnot’s work in 1824 [3], and named it in his honor; see [5].

***False Point 4.*** Some references also cite that Sadi Carnot stated that “the maximum cycle efficiency depends on the temperature difference of the two thermal reservoirs, implying it is a function of the temperature difference only [*η_C_* = *f(t_H_* − *T_L_)*]. However, it is misplaced, since Carnot’s statement was “in principle,” and he was fully aware that the maximum efficiency depends implicitly on the two temperatures only, but not their difference directly, as Carnot stated accurately [3]; see related Equation (6). 

Sadi Carnot reasoned and proved the reversible cycle, maximum efficiency based on logical “contradiction-impossibility” as emphasized next (see *Key Point* 18). He also detailed accurately how to accomplish the ideal cycles. The specifics and consequences of Sadi Carnot’s ingenious reasoning with his complete and accurate discoveries were presented in his, now famous publication [3] and elsewhere, including this author’s prior publications [16,17].

*Maximum efficiency* definition: The maximum thermodynamic efficiency of a work-producing process (or set of processes) is when “*maximum-possible work*” *is obtained* (or extracted), equal to the respective work potential (WP) of a system (or thermal reservoir) initial state (or input) with respect to its final state (or output), usually in equilibrium with a reference surrounding state; or when “*minimum-possible work” is supplied* to create the same initial state, with the same original WP from the same equilibrium state; i.e., when there is no work dissipation to heat of any kind during such reversible processes (or set of processes). 

The “*maximum cyclic-process efficiency”* is defined in the same manner since it consists of several reversible processes, and it expresses the WP of an energy-source system (thermal reservoir at higher temperature) per unit of energy (heat) consumed when reversibly interacting with a reference, energy-sink system (thermal reservoir at lower temperature).

The two, extracted and supplied works (or “work out” and “reverse-work into” a system), related to the same system non-equilibrium state and its respective equilibrium with the same reference surrounding system, must be maximum possible and minimum required, and both must be the same for all respective reversible processes, as a matter of *“contradiction impossibility*,” see *Key Point 18* and elsewhere [3,16,17].

***Key Point 18*.** Proof by “*contradiction-impossibility*” of an established fact is, by definition, the logical proof of the stated fact. If a contradiction of a fact is possible then that fact would be void and impossible. It is illogical, absurd, and impossible to have both, “the *one-way* and the *opposite-way*.” For example, if heat self-transfers from higher to lower temperature, it would be “*contradiction-impossibility*” to self-transfer in the opposite direction, from low to high temperature.

Otherwise, if a reversible process (including a cyclic process) with a smaller reversible, extraction work would be possible (with smaller efficiency than another reversible process), a part of the possible (original) WP would mysteriously vanish in an ideal process or cycle. Furthermore, its “reverse-process” (with its smaller work input would be more efficient), would then have a higher efficiency than the others, or with the others’ larger work, it could create a higher WP state than the original, as if such WP difference was a miraculous GAIN, created without due WP source; see Section 5.4.

Likewise, if the reverse process (with smaller reversible work) is coupled with another power process, to use its higher maximum work, it would result in spontaneous heat transfer from a lower to higher temperature (a contradiction impossibility of known fact). It would be equivalent to the generation of non-equilibrium from within equilibrium, or producing work from a single thermal reservoir, or the destruction of entropy; see relevant explanations with supporting *Figures* in [3,16,17].

Or, Sadi Carnot ingeniously reasoned that “such coupling of two different efficient reversible cycles would result in impossible creation of caloric.” Carnot’s reasoning methodology was ingenious and far reaching, and his final conclusions were accurate, regardless of his erroneous assumption of the conservation of caloric.

Therefore, if the heat is in fact always spontaneously transferred from a higher to lower temperature, then the spontaneous heat transfer in reverse direction, from a lower to higher temperature would not be possible, as the matter of contradiction impossibility. A similar process is true for any spontaneous energy displacement (energy transfer) against the respective energy force. A process spontaneity has a meaning for a process to be possible to proceed by itself (i.e., self-driven process “in self-physical, certain-direction, whatever it may be”) without any external influence or another external compensation. The external influence or compensation may be an external power process, or it may even be another internal, “conjugate power-process” or its tendency to drive as an external process, such as thermoelectrical phenomena, or thermomechanical or other elusive-like processes, influenced by gravity and other forced fields. Such mutually associated phenomena and processes may delude the “existence of miraculous processes,” resembling an impossible contradiction which they are actually not.

### 5.2. Carnot Cycle, Carnot Efficiency, and Carnot Equality (CtEq)

All related discoveries of the most important concepts of thermodynamics, after Carnot’s 1824 publication [3], regarding the reversible equivalency and maximum efficiency of reversible processes, in one way or another, were based on Carnot’s work. Namely, the absolute, thermodynamic temperature (not dependent on material of a thermometer nor its design; and not to be confused with the equivalent but not the same concept of ideal gas absolute temperature), the entropy concept, the Second Law of Thermodynamics, and the Gibbs free energy concept, among others. The *Carnot Equality* (CtEq) is specifically defined and named next (Figure 3 and elsewhere), to be distinguished from the well-known *Carnot Efficiency* (CtEf), the latter is also known as the *Carnot (Cycle) Theorem*.

**Figure 3 entropy-25-01106-f003:**
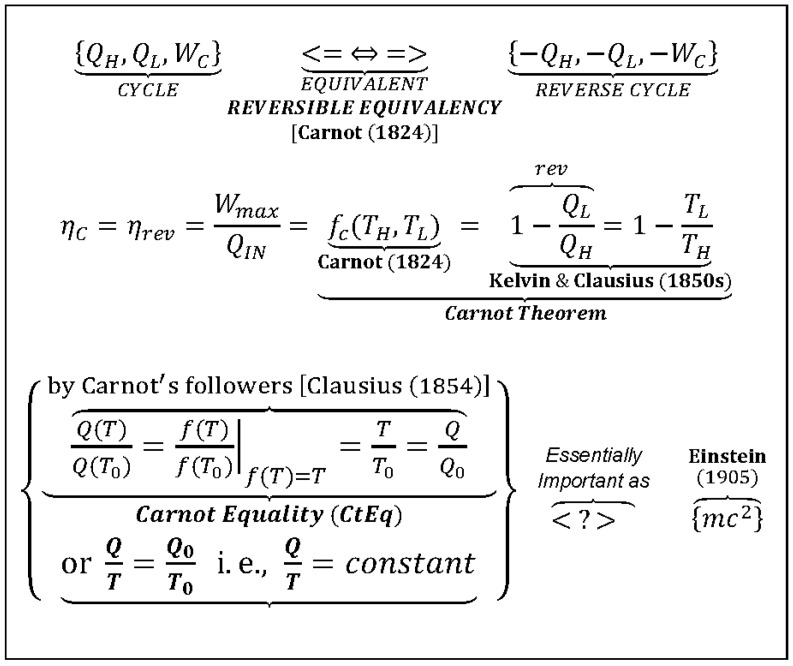
*Carnot Equality* (as named here), *Q/Q*_0_
*= T/T*_0_*,* or *Q/T* = *constant*, for reversible cycles (different from Carnot Theorem), is much more important than what it appears at first. It is probably the most important correlation in Thermodynamics and among the most important equations in natural sciences. Carnot’s ingenious reasoning unlocked the way (for Kelvin, Clausius, and others) for generalization of “thermodynamic reversibility,” definition of absolute thermodynamic temperature and a new thermodynamic property “entropy” (*Clausius Equality* is generalization of *Carnot Equality*), as well as the Gibbs free energy, one of the most important thermodynamic functions for characterization of electro-chemical systems and their equilibriums, resulting in formulation of the universal and far-reaching Second Law of Thermodynamics (2LT) (as originally stated by this author in 2008 [16] and 2011 [17]).

***Key Point 19.*** The *Carnot Equality* (*CtEq*), *Q/T* = *constant*, the well-known correlation, the precursor for the famous *Clausius Equality (CsEq)*, CI(*dQ/T*) = 0 (the cyclic integral for variable temperature reversible cycles), is specifically named here “as such” by this author in Carnot’s honor. The *CtEq* was based on Carnot’s 1824 discovery [3] that was finalized later by Kelvin (1850 using ideal gas) and generalized by Clausius (1854; see [5], pp. 69–109). The *CtEq* was also precursor for discovery of thermodynamic temperature and entropy. It is among the most important correlations in natural sciences, on par with Einstein’s famous, *E* = *mc**^2^*** correlation, see Figure 3 (as originally stated by this author in 2008 [16] and 2011 [17]).

***Key Point 20.*** The *Carnot Efficiency*, *CtEf*, *η_C_* = (1 − *T_L_/T_H_)*, a.k.a. *Carnot Theorem* (not to be confused with the *Carnot Equality*, *CtEq*) was originally established implicitly by Carnot, Equation (6), “as independent of cycle design and mode of operation,” therefore, in fact, not dependent on cycle *per se*, but *dependent on the thermal-reservoirs’ temperatures (T_H_ and T_L_) only*. Therefore, in fact, the *CtEf* represents the *WP* of the heat *Q_H_*, transferred from *T_H_*-reservoir while interacting with *T_L_*-reservoir only, i.e., it represents the *work potential of heat*, *WP_Q_* = (1 − *T_L_/T_H_)Q_H_*, realized by ideal, reversible Carnot cycle or any other, reversible steady-state device (so that any transient accumulation of heat or *WP* within the devices are excluded); see also *Key Point* 19.

Looking to reason the maximum possible efficiency of steam (heat) engines was a challenging mission, at the time when their efficiencies were below 5%, and neither nature of heat nor the concept of work from heat were known. Sadi Carnot mistakenly assumed, as many thought in his time, that heat was a weightless and indestructible caloric fluid. Indeed, the conservation of caloric was experimentally established within the calorimetric measurements. Furthermore, at that time, the difference between heat input and output in heat engines was negligible and within the experimental errors, due to extraction of a rather small work ratio (was only several percent of heat input)—so assuming the conservation of engines’ caloric appeared to be realistic at that time. Carnot’s reasoning (possibly and luckily instigated by the contradictory misconceptions at the time) that the heat engine concept has to be similar to the water wheels and that the “motive power” (work) is extracted by “falling [lowering] temperature of conserved-caloric [heat]”, the way the power has been obtained by lowering the elevation of the conserved water, falling through a water wheel; see [3], related contemporary discussions by this author [16,17], and elsewhere.

Consequently, Sadi Carnot reasoned that the maximum power efficiency has to be a function of temperature difference, although not directly linear (as sometimes misquoted), and that such “available temperature difference has to be the source of the motive power.” Then, he ingeniously concluded that any available temperature difference has to be utilized for increasing the amount of engine power and not to be “wasted” for heat transfer *per se*, since during the real-heat transfer at finite temperature difference no motive power (work) is extracted, and therefore, all work potential related to such temperature difference would be vanished. 

Therefore, to maximize a cycle efficiency, the heat transfer, in principle, has to be at “as little temperature difference as possible”, i.e., at an infinitesimally small difference and in isothermally limited at the temperature of the respective heat reservoirs. Then, Carnot reasoned that, to accomplish such isothermal heat transfer for the most efficient, ideal engine cycle, the working medium temperature has to be adjusted by frictionless adiabatic processes, for its temperature to be infinitesimally smaller than the high temperature of thermal source reservoir and infinitesimally higher than the low temperature of thermal sink reservoir, so that heat would be passing from the high to the low temperature between the thermal reservoirs with the assistance by the adiabatic processes, while the actual heat transfer would be reversible, at the infinitesimal temperature differences at each thermal reservoir. Therefore, the ideal Carnot cycle would comprise two isothermal and two perfect adiabatic (isentropic) processes; see Figure 4 (solid lines).

The reversible Carnot cycle also comprises isothermal processes where heat is entirely (100%) converted to work (*Q_H_ = W_H_*) while increasing volume and entropy (process 1–2), or in reverse, where work is entirely (100%) converted to heat (*W_L_ = Q_L_*) and there is decreasing volume and entropy (process 3–4). Note that the isothermal ideal gas heating is accompanied with the expansion work-out equal to heat-in, *W_H_* = *Q_H_*), while the quantity of its internal energy is unchanged. However, its quality is degraded (part of its work potential replaced with heat), as manifested by the increase in entropy (i.e., *U = constant*, but decrease in the WP and free energy *G = U* − *TS*)! 

In Figure 5, the conversion of heat and/or internal energy to work is presented in general, for a process from a “High-intensity Energy Source System (HESS, or H-reservoir)” at a higher temperature *T_H_*, to a “Low-intensity Energy Reference System (LERS or L-reservoir)” at a lower temperature *T_L_*, for an open or closed, steady-state or quasi-steady-cyclic process, respectively, including the irreversible loss of work potential to heat with entropy generation.

It is important to state that for any ideal reversible cycle, the reversible $Q_{L,R}$ is “*not a loss*” (as often incorrectly cited in the literature) but a *necessity* as demonstrated by the reversible Carnot Cycle (*Carnot Equality*, *CtEq*; see also *Carnot–Clausius Heat–Work Equivalency*, *CCHWE*, in Section 6). However, in real irreversible processes, unavoidable, so-called work “loss”, *W_LOSS_ = W_diss_ = Q_gen_ = T_L_S_gen_* (the Gouy–Stodola correlation), is due to the dissipation of work (*W_diss_*) into the generated heat (*Q_gen_*). Note that work as useful energy cannot be lost per se (1LT) but is dissipated, i.e., irreversibly converted to heat as a degraded form of energy. Additionally, for closed-mass and cyclic processes, *m_L_ = m_H_ =* 0, and for adiabatic turbine (*Q_H|L_ =* 0), *W**_OUT_** = E**_mH_** − E**_mL_***, see Figure 5.

***Key Point 21.*** During any *steady-state process* or *quasi-steady-cyclic process*, see Figure 5, the entropy input *S**_H_***, with heat *Q**_H_*** (and with mass *m**_H_*** if any) at *T**_H_*** >*T**_L_***, and any irreversible generated entropy *S**_gen_*** within, must be discharged with heat *Q**_L_*** (and with mass *m**_L_*** if any), as entropy *S**_L_*** at *T**_L_***.

Therefore, for a steady-state process in general (including the quasi-state cyclic processes), no transient accumulation of any property, ∂[…]/∂t = 0, a working system has to be “compensated thermally” (by transferring out the entropy supplied to the system, i.e., for a reversible cycle {*S_H_* = *Q_H_*/*T_H_*}*_IN_*= {*Q_L_*/*T_L_ = S_L_*}*_OUT_*, which demonstrates a logical proof of the *Carnot Equality*), and also to be “compensated mechanically” (by bringing a cyclic process to the initial volume), before repeating the cycle. Therefore, the heat rejected during a reversible cycle process, *Q_L,R_* = *T_L_*Δ*S_R_*, is the *necessity* and therefore “useful quantity”, not a loss as sometimes mispresented, see *False Point* 5.

***False Point 5.*** Citation in some references, that, “the heat rejected to the lower-temperature reservoir during a reversible cycle process, is a lost energy” is false, since it is necessary to remove the entropy input, in order to complete the cycle. Therefore, the rejected heat in a reversible cycle is the *necessity* and ‘useful quantity’, *not a loss* as mistakenly stated in some literature.

However, the irreversible “dissipation loss of WP” is unnecessary and should be minimized to increase efficiency. Therefore, the maximal cyclic work, *W_C_ = Q_H_* − *Q_L,R_ < Q_H_*, i.e., the Carnot cycle efficiency, *η_C_*, is always smaller than 100% but bigger than a real cycle efficiency, *η*, i.e.,
*η* = (*W_C_* − *W_LOSS_*)/*Q_H_* <{*η_C_* = *W_C_*/*Q_H_* = 1 − (*T_L_*/*T_H_*)} < 100%.(7)
Equation (7) represents the so-called 1LT energy conversion efficiency, not to be confused with the 2LT reversible efficiency, {*η_2LT_* = *η*/*η_C_*} ≤ {*η_C,2LT_*= 1 =100% = *η_R,2LT_*}. The curled term on the right of the inequality being the perfect 100% 2LT reversible efficiency for the Carnot cycle or any reversible process.

### 5.3. Carnot “Reverse-Cycle” and Thermodynamic “Reversible-Equivalency”

Furthermore, if the working medium temperatures in the Carnot cycle are adiabatically adjusted in reverse, to be infinitesimally higher than the high-temperature reservoir and infinitesimally lower than the low-temperature reservoir; see Figure 4 (dashed lines), then, the heat will be effectively-transferring in reverse, from low- to high-temperature reservoirs (from *T**_L_*** to *T**_H_***), while external work would be consumed. Therefore, all processes and energy flows would be in reverse direction, resulting in the “*Carnot reverse-cycle*” with regard to the original (power-producing) “*Carnot cycle*”, with infinitesimally different or in limit all equivalent, respective properties and energy flows, but in reverse directions; see Figure 4 and Equation (8). Such reverse cycles will provide cooling (refrigeration) of the low-temperature reservoir (any ambient; A/C or refrigeration cycle) and/or heating of the high-temperature reservoir (any ambient; heat-pump cycle), by effectively transferring heat from low to high temperature with utilization of external work.

Sadi Carnot [3] introduced the concept of reversible processes and cycles in 1824, as discussed above and elsewhere, and as expressed by Equations (6) and (8):(8)$$\underset{\mathrm{POWER}-\mathrm{CYCLE}}{\underbrace{\left\{Q_{H},Q_{L},W_{C}\right\}}}\underset{\begin{array}{c} MAY\;BE\;REVERESED\\ BACK-AND-FORTH\\ IN\;PERPETUITY \end{array}}{\underbrace{<\equiv >}}\;\underset{\mathrm{REVERSE}-\mathrm{CYCLE}}{\underbrace{\left\{-Q_{H},-Q_{L},-W_{C}\right\}}}$$

***Key Point 22.*** Sadi Carnot proved the *equivalency and maximum efficiency of reversible processes* by logically demonstrating that otherwise they will violate the contradiction impossibility of “logical criteria,” in his case, the *mistaken conservation-of-caloric criteria*; still it resulted in the correct conclusion due to the ingenious logic by Carnot. With rectified criteria and energy conservation, Carnot’s logic implied the *impossible self-transfer of non-conserved caloric from low to high temperature*, the contradiction of valid criteria used by Clausius.

We may reason an *alternate logical proof of “Reversible Equivalency”***:** Since ideal, *reversible processes* may effortlessly be self-reversed “*back-and-forth in perpetuity*” (Equation (8) and Figure 6), that imply they do not degrade their “energy quality,” and therefore, they have *maximum possible efficiency and are equivalent*—they are lossless or dissipationless. However, dissipative degradation of WP (energy quality) will diminish the WP and efficiency, and prevent “perpetual reversibility” or self-reversal.

The important concept of “thermodynamic *reversible equivalency*” is generalized and depicted in Figure 6 and presented elsewhere. The true equivalency, such as in frictionless mechanics, is conserved during ideal, reversible processes, but in real processes, the forced energy displacement towards equilibrium is accompanied by the useful energy or work potential degradation due to its irreversible dissipation to heat with entropy generation, as elaborated in Section 4.3 and elsewhere.

### 5.4. Work Potential, Formation Work, and Exergy

The formation (or creation) of non-equilibrium WP, requires the forcing of energy displacement on the expense of another non-equilibrium WP, where the non-equilibrium WP is displaced or transferred (along with entropy transfer with heat, if any), both ideally conserved (reversible equivalency), but in part or in whole, the WP dissipates to heat with irreversible entropy generation, at any space and time scale (for which ThD macro-properties are defined), without exception.

***Key Point 23.*** The *work potential (WP)* of a system state with regard to a reference state, is the unique, “energy quality” that could be reversibly retrieved as “*useful-energy*” if a system state is reversibly brought to a lower, reference equilibrium state, while interacting with respective reference surroundings. Such retrieved WP could be used in reverse as *formation work* to re-form the system state from that equilibrium to the original state, with ideal, reversible processes, thus defining the “*reversible equivalency*,” see Figure 6. Furthermore, if the “lower energy” state is chosen as a well-defined, standard *reference state*, then the WP becomes the “unique quantity” of such state, and hence, it may be considered as a system (quasi-) property, already defined as *exergy*. Some do not consider exergy as a property since it depends on the reference state, but that is also the case with some other properties. All WP-related quantities (*work potential, useful energy, formation work, exergy*), as asserted here and elsewhere, are directly interrelated and essentially the same concepts, they all irreversibly dissipate to heat in real processes, and they become zero at equilibrium.

***Key Point 24.*** Non-equilibrium, useful energy or WP is directionally transferred (from higher to lower energy density) and conserved in ideal reversible processes (1LT), while in real processes the WP is irreversibly dissipated (converted) into heat with entropy generation (2LT); however, conserved “as work-and-heat” in totality (1LT), as detailed elsewhere.

## 6. Thermal Transformers: Carnot–Clausius Heat–Work Reversible Equivalency (CCHWRE)

### 6.1. Thermal Transformers and CCHWRE

*Thermal transformers* were named and discussed by this author in 2004 [12] and revisited later [17]. See also Appendix B.

***Key Point 25.*** Heat transfer (“thermal poking”) requires a higher temperature source and is always accompanied by entropy transfer (*δQ = TdS*). However, if an energy is transferred in an orderly manner, without entropy transfer, then it is not heat but adiabatic *work transfer—it increases energy and temperature but is ideally isentropic,* without entropy transfer. *That is how the temperature could be increased without heating*. In reverse, adiabatically extracting work would lower temperature without cooling, otherwise, the latter would require a lower temperature heat sink — namely, *only work could increase temperature above or decrease below the available source or sink temperatures, respectively*.

***Key Point 26.*** *Thermal transformers:* With all relevant processes, working in sequence as the cycle, the reversible heat transfer from any to any temperature level could be achieved, functioning as a “*reversible thermal-transformer*.” Namely, the reversible heat transfer from higher *T_H_* to lower *T_L_* temperature with *W*, Carnot cycle work output; or in reverse, the reversible heat transfer from lower *T_L_* to higher *T_H_* temperature with *W*, Carnot cycle work input. Likewise, the real *thermal transformers*, as combined power heating and refrigeration cycles (including heat-pump cycles) also transfer heat from any to any temperature level, except with reduced efficiency due to the unavoidable dissipation of WP into generate heat and entropy (Equations (5) and (7)).

*Carnot–Clausius Heat–Work Reversible-Equivalency* (*CCHWRE*), as named and “*enlightened”* here, establishes specific *equivalency* (or specific “*interchangeability*”) between *heat and work*, as per Figure 7 and Equations (9) and (10). It is based on the early work of Carnot (1824) [3], that “all reversible processes and cycles have equal and maximum efficiency for the given thermal reservoirs temperatures, regardless of device and mode of operation,” and among others, including Thomson (Kelvin) and Clausius’ meticulous work, around 1850s [4,5].

Clausius “struggled,” in his “*Mechanical Theory of Heat*” [5] (Ch. IV: *Principle of the Equivalence of Transformations*), to fully decern the Carnot’s postulates, and to finalize his ‘*transformations’,* i.e., “when heat is reversibly transferring from high temperature and in part releasing [converting to] work, and in part transferring to heat at low temperature”.

Clausius’ reasoning was as ingenious as Carnot’s, with debatable particulars, but with accurate final deductions of the “*transformations*” equivalence-values,” with the *f*(*t*) *=* 1/*T* integration factor, that resulted in the definition of new property, the *entropy* (the “missing transformation”) and definition of the quantitative correlation of the Second Law of Thermodynamics (2LT), including the *Clausius Equality* (CsEq) for reversible cycles, and *Clausius Inequality* for all real, irreversible cycles, the latter include entropy generation caused by work dissipation to heat, due to irreversibilities of different kinds (see also Section 4.5).

The CCHWRE is demonstrated by the Carnot cycle (and other power cycles along with the reverse-refrigeration cycles), where the Carnot cyclic-work (*W_C_*) and rejected heat at lower temperature (*Q_L_*) are obtained from heat at higher temperature (*Q_H_*) alone; and in-reverse, where the utilized work in refrigeration reverse cycles (*W_C_*) is added (in) to a heat at low temperature (*Q_L_*), resulting in the heat at higher temperature (*Q_H_*) alone, see Figure 7 and Equations (9) and (10).

Due to reversible equivalency, as originally devised by Carnot [3] (see *Key Point* 22 and Equation (8)), elaborated by Clausius [5], and re-interpreted in Section 5 (Equation (8) and Figure 6), it is evident that heat at different temperatures and work is not the same energy quality. However, they are (reversibly) interrelated (they morph into each other, as emphasized on Figure 7 and Equations (9) and (10)), and in that regard they are interrelated or “interchangeable” as follows:(9)$${\;Q}_{H}\equiv \underset{reversible}{\underbrace{{\;W}_{C}+Q_{L}}}\;\;or\;\;W_{C}\equiv \underset{reversible}{\underbrace{Q_{H}-Q_{L}}}\;or{\;Q}_{L}\equiv \underset{reversible}{\;\underbrace{{Q_{H}-W}_{C}}}$$
(10)$$\frac{Q_{H}}{T_{H}}=\frac{Q_{L}}{T_{L}}=\frac{W_{C}}{T_{H}-T_{L}}$$

The above correlations are much more important than they appear at first, since they represent the “heat-work true reversible-equivalency” in general, for all reversible steady-state processes not only for cycles (see Figure 7). Namely, heat *Q_H_* at high temperature *T_H_* is equivalent with the sum of heat *Q_L_* at lower temperature *T_L_* and Carnot’s work *W_C_*, Equation (9) *Left*; or any other relevant rearrangement, Equation (9) *Center* or *Right*, along with the reversible *Carnot Equality*, as formalized and named here, Figure 3 and Equation (10). 

The above correlations, Equations (9) and (10), the latter in simple arithmetic form for constant temperature of the thermal reservoirs, will require proper integration for thermal systems with variable temperatures, the way the *Carnot Equality* (named and highlighted here, see Section 5.2), is generalized with the integral form in the *Clausius Equality* [5].

### 6.2. Proof of Ideal Gas Equation of State and CCHWE Confirmation

*Ideal gas equation of state:* Ideal gas (IG) is composed of many randomly moving, hypothetical massive point-particles that undergo elastic collisions but without any other particle interactions. Regardless of its simple structure, the IG is a good approximation of the behavior of many real gases in many applications. The IG energy consists of random- or *thermal motion* (ThM) of its particles. It may be expressed as its thermal energy *[E_IG_* ≡ *E_ThM_]* =***E_th_** = N(k_B_T) = nR_u_T*, where, *N* is number of particles, *k_B_* [*J*/*K*] is the Boltzmann constant (or energy-temperature conversion factor), and *T* is particle-average absolute temperature, i.e., *k_B_T* is the particle average energy, *n* is number of moles, and *R_u_* is the universal, molar gas constant.

All systems allow the storage and transfer of ubiquitous thermal energy (since the ThM cannot be averted, even absolute-0K cannot be achieved, the 3LT); however, the rigid systems do not allow storage of mechanical compression energy like gases. More complex systems with relevant structures may allow the storage and transfer of other energy types, such as electrical charging, magnetization, chemical or nuclear reactions, and similar.
(11)$$\mathrm{Duality of random (thermal) motion energy:}\;\mathrm{Mechanical}\equiv \mathrm{Thermal}\;\underset{E_{me}}{\underbrace{P\cdot V}}\equiv \underset{E_{th}}{\underbrace{n{\cdot R}_{u}\cdot T}}$$

The ThM of IG particles along with temperature also exhibit the pressure on any hypothetical or real boundary surface and, therefore, its energy may also be represented as mechanical (pressure) energy: [*E_IG_* ≡ *E_ThM_*] *= **E_me_*** = *PV*, where *P* is mechanical pressure (defined as relevant energy per unit of volume), and *V* volume of IG. Therefore, we may express the *IG equation of state* (i.e., the constitutive correlation of its mechanical and thermal properties), as the equivalence (“**≡**”) of the two forms of the same energy (Equation (11)).

***Key Point 27.*** The *reasoning here presents a logical proof of the IG equation of state*. The duality of manifestation of IG’s ThM energy, either as mechanical (via pressure and volume) or as thermal (via temperature of particles), demonstrate why the IG structure (random ThM of its particles) enables interchangeability of heat or work storage and transfer, depending if energy is stored or transferred via thermal-motion (ThM), by “jiggling” across a boundary surface (at constant volume) and thus changing the temperature and entropy, or by mechanical displacement of the boundary and changing the pressure and volume.

However, due to the duality of the ThM, the *P* and *T* are conjugate and interrelated via the IG equation of state, Equation (11). Similarly, real gases (including steam, called simple (thermo-) compressible substances) allow thermal and mechanical energy storage and transfer, and manifest the duality and interchangeability of heat and work, as formulated and named here; see next.

*Carnot–Clausius Heat–Work Reversible Equivalency* (CCHWRE) *confirmation*: The CCHWRE will be demonstrated and confirmed using IG for its simplicity. Furthermore, since the correlations for the *reversible equivalency* are, in principle, valid in general, regardless of intermediary, working system or mode of its operation (any reversible process or cyclic path are equivalent), the results obtained with IG will be, in principle, valid in general.

The reversible Carnot cycle (see Figure 8) also comprises isothermal processes where heat is entirely (100%) converted to work (*Q_H_ = W_H_*) while increasing volume and entropy (process 1–2), or in reverse, where work is entirely (100%) converted to heat (*W_L_ = Q_L_*) while decreasing volume and entropy (process 3–4). Note that the isothermal ideal gas heating is accompanied with the expansion work-out equal to heat-in, *W_H_* = *Q_H_*), while the quantity of its internal energy is unchanged. However, its quality is degraded (part of its work potential replaced with heat), as manifested by the increase in entropy (i.e., *U = constant*, but decrease in the WP and free energy *G = U* − *TS*). Furthermore, the Carnot cycle comprises reversible adiabatic processes where internal energy (stored heat in IG) is entirely (100%) converted to work, −*Cv*(*T_L_* − *T_H_*) *= W_23_*, by lowering temperature and increasing volume (process 2–3), or in reverse, where work is entirely (100%) reversibly converted to internal energy (stored heat in IG), *W_41_ = Cv*(*T_H_* − *T_L_*), by increasing temperature and decreasing volume (process 4–1 on Figure 8).

Note that the reversible, adiabatic works have equal magnitudes (|*W_23_*| = |*W_41_*|) and cancel out for the whole cycle (they have the main purpose to change temperature levels for the reversible heat transfer to be possible). Consequently, the net cycle work is the result of the isothermal works’ difference due to the respective temperature difference, *W = |W_H_|* − *|W_L_| =* (*T_H_* − *T_L_*)|*ΔS*|, while exchanging the same entropy in and out, |*ΔS_12_*| = |*ΔS_34_*| = |*ΔS*|, so that entropy cancels out, enabling the completion of the cycle (note that the isentropic works do not contribute to the entropy balance). For more details and all the specific equations, see Table 1 in [17] (p. 344).

### 6.3. Reversible Cycles Are 100% Efficient and Carnot Efficiency Is Essentially the “Measure” of Heat-WP

The following *Key Points* are articulated to further emphasize and summarize relevant facts and consequences of the “thermodynamic reversible equivalency.”

***Key Point 28*.** *All reversible processes (including cyclic processes) under the same conditions must have equal and maximum efficiency, as demonstrated by relevant “contradiction impossibility.”* As a matter of fact, the reversible processes and cycles were a priori “specified” as ideal, with maximum possible efficiency, with a priory 100% 2LT reversible efficiency, not dependent on their design or mode of operation (independent of their quasi-stationary cyclic path or any other, reversible stationary process path). Actually, as the ideal ‘*work-extraction measuring-devices*’, all reversible processes and cycles, in fact, determine the WP (as % or ratio efficiency with reference to relevant total energy) of an energy-source system with another reference system (such as with the two thermal reservoirs with the Carnot cycle, so their WP ratio is dependent on their temperatures only).

***Key Point 29*.** The maximum-possible *work potentia*l (WP) of a system (thermal-reservoir or any other), between any two states (its initial and final (reference) states), is *independent of the process path* or the process device properties or design (cyclic or otherwise), that reversibly brings the initial energy state to another reference state (by reversibly interacting with reference surroundings towards an equilibrium state), but it *only depends on the two states’ relevant properties*, e.g., it only depends on the temperatures of the two thermal energy reservoirs (as ingeniously deduced by Sadi Carnot in 1824 [3]).

***Key Point 30.*** The *ideal heat-engine cycle* is just a (cyclic) process path between high- and low-temperature thermal reservoirs, and the maximum possible *reversible efficiency* is not dependent on the cycle device and process path, but *only dependent on the two temperatures* (as originally reasoned by Sadi Carnot in 1824 [3]), and elsewhere, including by this author [16,17]. The reversible cycles are “used” to evaluate (“measure”) mutual, maximum efficiency of the thermal reservoirs, not the cycle *per se*, since it is independent of the cycle design and mode of operation, therefore not dependent on the cycle *per se*, as implicitly postulated by Carnot, “maximum work is obtained by any reversible cycle, independent form the medium used or mode of operation, is *dependent only on the temperatures of the two heat reservoirs* [hence, *not dependent* on the cycle but the reservoirs’ properties/temperatures only].”

***Key Point 31.*** The cycles are only intermediary devices, such as different “paths of operations” and all deductions and correlations derived refer to “the heat from the high temperature reservoir being transformed [i.e., converted] to “extracted work and remaining heat transferred to the lower temperature reservoir”; and in reverse, with all relevant quantities having equal magnitude in opposite directions. The Carnot work refers not only to thermal cycles but also to the thermoelectric and *other steady-state devices*, i.e., it refers to thermal work potential of an energy source in general. This rationalization will require further elucidation in separate writings.

As stated before, the reversible Carnot cycle is the “measuring yardstick” of related equivalencies of all relevant input and output quantities, see Figure 4 and Figure 8).

## 7. “*No Hope*” for the Challengers of the Second Law of Thermodynamics

“*It is hard to believe that a serious scientist nowadays, who truly comprehends the Second Law and its essence, would challenge it based on incomplete and elusive facts […] However, sometimes, even highly accomplished scientists in their fields do not fully realize the essence of the Second Law of thermodynamics* [16,17,18,19,20,21,22,23].”

As already stated, this treatise [1], written for a special occasion [2], is presenting this author’s lifelong endeavors and reflections [12,13,16,17,18,19,20,21,22,23], including additional, original reasoning and interpretations, regarding the fundamental issues of thermodynamics, and especially as related to the subtle Second Law of Thermodynamics (2LT), as well as to put certain physical and philosophical concepts in historical and contemporary perspective.

In this section, a number of related issues are presented and emphasized with several *Key Points*, including a *Deceptive Example* (Section 7.1), “Three *Primary-deception structures*” of the 2LT, classified by this author (Section 7.2), and critical discussions on the two selected publications by avid challengers of the 2LT, one recent publication, challenging the 2LT [24], and another, self-claimed as a “landmark paper”, experimentally challenging the validity of the 2LT [25]; see Section 7.3 below.

A small, but adventurous and stanch group of creative scientists and inventors-to-be, named here as the 2LT “*Challengers*,” are “bravely” challenging the 2LT universal validity, often based on a fact that they have been successful in achieving a perpetual non-equilibrium (with limited work potential, WP) but without perpetual work production, using innovative and creative methods and processes, and hoping to utilize it to “somehow” perpetually produce work (useful energy) from within the environment alone, as a single thermodynamic reservoir in equilibrium.

***Key Point 32.*** The current frenzy about violation of the 2LT, of *getting “useful energy” from within equilibrium alone* (with the “Perpetual-motion machine of the second kind”, PMM2), is in many ways similar, but more elusive and opportunistic, than the prior frenzy about violation of the First Law of Thermodynamics (1LT), of *obtaining “useful-energy” from nowhere* (with the “Perpetual-motion machine of the first kind”, PMM1).

However, nobody has been successful to achieve spontaneous and sustained conversion (stationary or cyclic) of the surroundings’ (thermal) energy to useful work, nor to provide reliable evidence (comprehensive energy and entropy “accounting”) of achieving a sustainable, overall process efficiency higher than the Carnot’s maximum possible (which is zero from a single thermal reservoir only).

***Key Point 33.*** The driving force of any process (or change) is the non-equilibrium or useful energy (or WP, or related free energy, or *exergy*) that exhibit a forced directional tendency for its displacement towards mutual equilibrium and not in the opposite direction (i.e., “energy ability to do work [and transfer heat]” or *to produce change*). *It is illogical and pointless “to insist on the impossible-possibility”* for the self-producing non-equilibrium from within the equilibrium alone without required forcing. A new WP cannot be created since it only can be displaced (or transferred), and it is always diminished due to its irreversible dissipation (or its conversion to heat with entropy generation) until mutual equilibrium with uniform properties and maximum entropy, is asymptotically achieved—a “dead-state” without WP required for any further change. Even the heat transfer at finite temperature difference is caused by its Carnot thermal WP (“heat exergy”).

### 7.1. “Perpetual-Motion Watch” Deceptive Example

We may “wishfully hypothesize” miraculous processes to achieve impossible outcomes by overlooking elusive but critically important phenomena. We will present here one trivial example that may appear to be a *“perpetual-motion watch”*, in order to demonstrate that without full knowledge what is inside a “black-box”, it may deceive some to “jump to unjustified conclusions” without due comprehension.

If a watch is running for years without supplying any energy from outside, it may appear without knowing what is inside the watch, that it somehow runs by self-creating energy (PMM1) or that it somehow produces useful-energy from the surroundings as a single thermal reservoir (PMM2, self-creating WP), see Figure 9, with a pictured real watch with 5–10 years battery life.

It would be wrong to hypothesize that the useful work is perpetually supplied by heat from the surroundings alone (a violation of the 2LT), or that the work is somehow miraculously generated from nowhere (a violation of the 1LT). 

Similarly, many other subtle electro-chemical and other interactions at different time and space scales may allude to a violation of the fundamental laws, especially for near-reversible processes with negligible frictional and other dissipations, resembling “perpetually self-running watch” on Figure 9, running for years without any supply of energy from the outside. Some near-ideal processes, with apparent perpetual motions, may appear to work without energy consumption or be “mistakenly hypothesized” to somehow, spontaneously produce work from the surrounding equilibrium only. 

Since the 1LT of energy conservation appears more intuitive and the 2LT is more elusive, the inventors-to-be or even some seasoned researchers may mistakenly hypothesize the spectacular inventions that violate the 2LT. We should be very careful to avoid premature and sensational hypotheses based on inadequate experiments and incomplete analyses.

### 7.2. Three “Primary Deception Structures” of Hypothetical Violation of the Second Law of Thermodynamics

(1) *First-deception structures* (or “*dynamic [quasi-] equilibrium*”) are about confusing the notion of “*perpetual free-motion (or free-oscillation)*” without load (without extracting useful energy but self-sustaining unavoidable dissipation), with notion of “*perpetual motion machine*” of perpetually producing useful work without due WP source. It was named “*dynamic [quasi-] equilibrium*” and it has been discussed in more detail by this author in [17,18,19,20,21,22], see Appendix B and elsewhere.

(2) *Second-deception structures* (or “*structural [quasi-] equilibrium*”) is about creation of “systems with perpetual non-equilibrium properties” with transient, limited WP (such as non-uniform temperature or pressure, or EM charge, and similar), to be somehow miraculously utilized for “perpetual self-creation of useful-work” from within an equilibrium surroundings alone, thus without due, perpetual WP source [25]; see Section 7.3. The former, a perpetual “non-equilibrium system state,” with limited WP energy, is not at all the same as a “self-creation of perpetual work” from within an equilibrium, or having a more efficient perpetual cycle than the Carnot cycle. It would be against the forcing direction of energy displacement, from higher to lower energy density (it would be a PPM2); and it would defy the very existence of equilibrium (it would be the equilibrium’s “contradiction impossibility”). It is called “*structural [quasi-] equilibrium*” and it has been discussed in more detail by this author in [17,20,21,22] and elsewhere.

(3) *Third-deception structures* (or “*persistent-currents quasi-equilibrium*”) is about certain “persistent currents” phenomena with self-perpetual, such as *“dissipate-and-reverse”* processes (including the Brownian motion, and the Meisner effect, as a process reverse to ((micro) irreversible) dissipation of the “superconducting persistent currents” or non-superconducting “persistent currents in metal rings” [24], and similar), that appear to perpetually micro-dissipate and reverse WP within their locality in equilibrium, as if they “quasi-reversibly dissipate-to-themselves” or self-create WP for its own dissipation; and thus, as if they violate universal validity of the 2LT, regardless of not producing any useful energy.

The challengers argue that such phenomena, in principle, “disapprove” the 2LT universal validity, and could potentially, with future innovations, be used as PMM2 to produce useful-energy while violating the 2LT [24,25] (with large number of references therewith). However, the existence of such structures in quasi-equilibrium, since they do not produce perpetual work from within an equilibrium, is not justification for valid violation of the 2LT.

***Key Point 34.*** In fact, any process that perpetually self-sustain its own macro-structure, regardless of whether it is with uniform or non-uniform macro-properties, is in equilibrium or quasi-equilibrium, respectively; and it is “in its own right”, reversible (perpetually self-sustained). Furthermore, as a matter of logic, the *reverse-process perpetuity* implies *maximum efficiency and reversible equivalency*—it is the required condition and definition of reversibility. That is why the reversible processes are called quasi-static or quasi-equilibrium processes; see next *Key Point*.

***Key Point 35.*** The *ideal gas* (IG) micro-structure consists of chaotic ThM of perfectly-elastic particles, and in equilibrium, their *collisions are reversible* and without dissipation (as if the ThM “micro-dissipates to itself”). However, during an adiabatic free expansion (no heat nor work transfer), its energy will not change but entropy will be *irreversibly* generated due to its volume increase, regardless that the ThM collisions are elastic. Furthermore, if during a phase change of a system, its expansion is isothermal–isobaric while in *equilibrium* with its surroundings, such process will be *reversible*: work-out (to displace surrounding) would be equal to heat-in from the surroundings, and without entropy generation (its entropy increase will equal to entropy decrease in the surroundings, or vice-versa).

Therefore, the quasi-static (or quasi-equilibrium) *phase change* of real systems in equilibrium with its surroundings are ideally reversible processes and could perpetually be reversed back and forth with an infinitesimal change in respective intensive properties, i.e., they are *virtually irreversible* within their self-sustained, perpetual *virtual equilibrium*. Similarly, the *persistent-currents equilibrium* (as the *Third-deception structures*) may be *virtually irreversible* (or *near-reversible*) within their self-sustained and perpetual, *virtual (quasi-) structural equilibrium*. Such structures do not self-produce perpetual work nor perpetually destroy entropy, and therefore do not violate the 2LT, as speculated in [24,25].

***Key Point 36.*** Hypothesizing a violation of the 2LT, or worse, claiming that the existence of some unexplained structures or phenomena, “disapprove” universal validity of the 2LT, is misplaced and impossible since it would be the contradiction impossibility of the proven reversible equivalency: *if the 2LT is not valid in a particular case, then it would not be valid in general due to reversible equivalency* [17,18,19,20,21,22,23] (and elsewhere).

***Key Point 37*.** Real thermal motion (ThM) with accompanying collisions (not perfectly elastic such as IG collisions), evidently “micro-dissipates-to-itself”, and the ThM is self-sustaining the perpetual, thermal *macro-equilibrium*; therefore, it is a *reversible phenomenon*. Similarly, the Brownian motion or chemical equilibrium reactions or electro-magnetic currents or any self-sustained quasi-equilibrium local phenomena, appears to be micro-dissipating into and are driven in reverse by the surrounding ThM (including thermal EM radiation) or are negligibly irreversible; and if in self-sustaining quasi-equilibrium, they have to be *virtually macro-reversible*.

***Key Point 38:*** During the WP transfer and conversion/storage, a part of *WP will always* and everywhere, *without exception, dissipate to heat and generate entropy*, until all WP is ultimately dissipated, being zero at *ultimate equilibrium*. However, in the process towards ultimate equilibrium with no work potential (“*thermodynamic death*”), the new structural, temporary, and localized quasi-equilibriums may be established and self-sustained within certain bounded structures, with residual work potential related to their surroundings. Every such *quasi-equilibrium* is represented with self-sustained micro-fluctuations, or micro-perpetual motions, including ultimate thermodynamic equilibrium (e.g., residual cosmic radiation).

***Key Point 39.*** Many creative hypotheses of *wishful-inventions*, to create useful energy from within the surrounding equilibrium, against the natural forcing, *have never materialized*, since it would be the “contradiction-impossibility” of existence of self-sustained stable equilibrium and natural forcing of non-equilibrium energy displacement towards mutual equilibrium of all interacting systems.

***Key Point 40.*** Therefore, the spontaneous displacement of energy from lower to higher energy density, in opposite direction from natural forcing and self-creation of non-equilibrium, would be similar to forcing in one direction with acceleration in opposite direction. Such wishful thinking would be the natural contradiction impossibility. It would negate stable equilibrium existence and will imply self-creation of WP with entropy destruction. Consequently, a non-equilibrium, the source of WP and forcing, cannot be generated (contradiction impossibility of unavoidable dissipation), but could only be transferred and ideally conserved, while in realty, a WP will tend to dissipate to heat (within a complex micro-structure and fluctuating micro-processes), towards a mutual macro equilibrium.

As the fundamental laws of nature and thermodynamics are expended from simple systems in physics and chemistry, to different space and time scales and to much more complex systems in biology, life, and intelligent processes, there are more challenges to be comprehended and understood. There is a need to better discern the fundamental concepts at different scales and complex systems, such as diverse and more complex and self-sustained “*structural equilibriums*,” without net-fluxes at scale of interest, but with non-uniform concentration potentials under coupled force fields; e.g., hydrostatic pressure and adiabatic temperature distributions in gravity field, charge, and species concentration distributions in electromagnetic or chemical fields, as well as to differentiate between transient and “near stationary” processes under influence of known and stealthy boundary and field conditions.

### 7.3. “Experimental Test of a Thermodynamic Paradox” Demystified

Professor Sheehan, an avid 2LT challenger, who organized several 2LT conferences and wrote extensively regarding the validity and violations of the 2LT, was claiming in his landmark paper with colleagues [25], also in references therewith that, *“There are now roughly three dozen theoretical proposals for its violation in the mainstream scientific literature, more than half of which have resisted resolution as of this date. There are also experiments which purport to violate the 2LT, and not all of these have been discounted. My experiments in particular, published in Found. Physics in 2014* [25]*, have not been disproved or shown to be in error in any meaningful way.”*

It is reasoned and argued here that the claims in the landmark paper by Sheehan et al. [25] are misplaced and over-stretching. Even if “*more than half*” of the challenges “*have resisted resolution as of today*”, the other half have been disavowed and none have been verified to date. Especially problematic are incomplete, misleading, and biased experimental results, as if the challengers are not comprehending or “conveniently ignoring” the very fundamentals and essence of the 2LT.

Specifically, the last two concluding sentences of Sheehan’s et al. paper [25] were, *“In summary, Duncan’s temperature difference has been experimentally measured via differential hydrogen dissociation on tungsten and rhenium surfaces under high temperature blackbody cavity conditions. We know of no credible way to reconcile these results with standard interpretations of the second law.”* The claim of *“black-body cavity conditions”* is questionable since it should be of much larger size than the devices inside. However, even if within a black-body cavity, the existence of stationary, non-uniform properties (non-uniform temperatures, etc.) will not violate the 2LT of thermodynamics. 

The last sentence of the paper is rather speculative, since the paper results describe a non-homogeneous, structural equilibrium established after externally imposed non-equilibrium (by heating the container tube to a very high temperatures). However, the 2LT, is classically stated for simple compressible substances for heat–work interactions only (where temperature is uniform at equilibrium). In general, the 2LT describes process conditions during the spontaneous directional displacement of mass energy (cyclic or stationary extraction of work), accompanied with the irreversible generation (production) of entropy due to the partial dissipation of work potential to thermal heat, which was not tested at all by the reported experiments, but only hypothetical and wishful claims were stated. After all, before the 2LT-violation claims are stated, the reliable criteria for the 2LT violation, including *proper definition and evaluation of entropy balances* (very important), should be established based on full comprehension of the fundamental laws of nature.

Even more problematic is the authors’ claim that their experiments “*point to physics beyond the traditional understanding of the second law*,” to justify their belief regarding the possibility of the 2LT violation, without due clarification and justification. The experiments relate to a special system with non-uniform temperature distribution due to dissociation/recombination, but do not represent a black-body cavity, and especially do not relate to the essence of the 2LT verification nor violation, as detailed below:1.Most of the fundamental formulations of the phenomenological, classical thermodynamics (called “*standard thermodynamics*” in the last sentence in Section 2 in the paper [25]), are reasoned and derived for the “simple compressible thermodynamic system,” the latter structure allows for heat and mechanical work interactions and storage only, but not other interactions, as well as for the ideal, black-body cavities, with uniform thermodynamic properties in equilibrium (uniform temperatures and pressures in such simple material structures and systems). The experimental system described in the paper is not nearly closed to the ideal black-body cavity, and the described, dissociation/recombination interactions between heterogeneous devices within a controlled isothermal tube (of the same order of magnitude size as the devices inside) are much more complex than a simple thermo-mechanical interaction of a simple compressible system in an ideal black-body cavity.2.The stationary quasi-equilibriums (with non-uniform properties) are abundant in nature, and do not violate the 2LT at all. For example, hydrostatic pressure distribution in a container, or adiabatic atmospheric temperature distribution, or non-uniform distribution of other properties in a stationary equilibrium, in gravity, electromagnetic or chemical fields, such as the presented results. I called the above a “structural equilibrium” (sustainable equilibrium but with non-uniform properties), as opposed to ideal thermodynamic equilibrium (with uniform properties) between the simple compressible systems with boundary heat and work interactions only, immune from any other structure or field interactions. This is one of several other problems of the paper’s judgments and conclusions.3.The statements in Section 6 show a discussion in paper [25], *“Within the traditional understanding of the second law, stationary temperature differentials such as those reported should not be possible.*” This statement is arbitrary and not justified; see also comments above. Likewise, *“Second, the temperature differences in DP experiments generated Seebeck voltages that can drive currents—and did, through their thermocouple gauges—thus, were capable of performing work like a heat engine.*” This is pure speculation, since we do not know what kind of stationary process will re-establish if a heat engine (HE) or electrical load is interfaced to utilize temperature or Seebeck voltage differences within the described system and devices.
A simple question arises: *why have the authors not experimentally verified their hypothesis*, if a stationary work extraction would be possible from within an environment in equilibrium? Such straightforward experiments could and should have been performed to experimentally check such a critical hypothesis. Based on classical thermodynamics, which allows transient processes, after an initial non-equilibrium is externally imposed (as in the paper experiments), the appropriate stationary structural equilibrium with property gradients will establish, as in the paper, a stationary process with perpetual work extraction outside of an equilibrium which is not possible, without external perpetual work source.
4.The last two concluding sentences of the paper were “In summary, Duncan’s temperature difference has been experimentally measured via differential hydrogen dissociation on tungsten and rhenium surfaces under high temperature blackbody cavity conditions. We know of no credible way to reconcile these results with standard interpretations of the second law.” The assumptions and conclusions are misleading and unjustified, as specifically described above.

The authors [24,25] (and a number of other “*Challengers*” of the 2LT) often misinterpret the fundamental laws, present elusive hypotheses, and perform incomplete, biased experiments, always short of simple confirmation of their 2LT violation claims. The authors’ implication that with creative devices, “*Challengers’ Demons*” (like *Maxwell Demon;* see Ref. [22]), it is possible to imbed them to a macro-equilibrium environment and extract stationary “useful work”, are philosophically and scientifically unsound. Such magic and wishful “demons”, if possible and inserted as a “black box” in a system or environment at equilibrium, to create a steady-state (stationary) work-extracting process from within such equilibrium, would be in opposite direction of existing natural forces, and also be a “catastrophically unstable” process with a potential to “syphon” all existing mass energy in an infinitesimal-size singularity with infinite mass-energy potential, which is super black-hole like. If it were ever possible, we would not exist “as we know it” here and now!

## 8. Conclusions

As already stated, this comprehensive treatise [1], written for a special occasion [2], presents this author’s lifelong endeavors and reflections [12,13,16,17,18,19,20,21,22,23], including original reasoning and re-interpretations, regarding the fundamental issues of thermodynamics, and especially as related to the subtle Second Law of Thermodynamics (2LT), as well as to put certain physical and philosophical concepts in historical and contemporary perspective; see Section 1 (*Introduction,* which ends with the *Selected Abbreviations and Notes*). The main content of this treatise was presented in the following *Sections*:

In Section 2, “*Energy forcing and displacement,*” the related concepts have been pondered. The “force or forcing” is non-equilibrium energy tendency to displace or redistribute (or to extend) from its higher to lower energy density (or energy intensity) towards mutual equilibrium with uniform properties. Typical, energy *intensive* and *extensive* conjugate properties (*energy force* and *energy displacement*) were presented in Table 1. All but thermal energy displacements are conserved, while *thermal displacement* (*entropy* or number of *thermal virtual particles*, *N_ThVP_*, as defined and named here) is irreversibly generated due to the dissipation of all other energy types to heat.

Then, in Section 3, “*Reasoning logical-proof of the fundamental laws*,” the concept of energy forced displacement as the mechanistic phenomenon in general was reaffirmed. The elementary particles (including “field-equivalent” particles) or bulk systems (consisting of elementary particles), mutually interact along shared displacement (with equal, respective action–reaction forces), thereby conserving energy during their interactive, mutual displacement. Since all existence is in principle mechanistic and physical, it was demonstrated here that the *Laws* of Thermodynamics (LT) are generalized extensions of the fundamental Newton’s Laws (NL) of mechanics. The First Law of Thermodynamics (1LT) is the generalized law of the conservation of energy, and the Second Law of Thermodynamics (2LT) describes the forcing tendency of non-equilibrium, useful energy (or work potential, WP) for its displacement and irreversible dissipation to heat with entropy generation, towards mutual equilibrium.

In Section 4, “*Ubiquity of thermal motion and heat, thermal roughness, and indestructability of entropy*”, this author’s comprehension of related phenomena has been further advanced by defining a new concept of “*thermal roughness*” and reasoning impossibility of entropy destruction, among others. Entropy, as the “final transformation” cannot be converted to anything else nor annihilated, but only transferred with heat and irreversibly generated with heat generation due to work dissipation, including Carnot “thermal work-potential” dissipation. 

The “*thermal roughness*” and related “*thermal friction*” were defined and named here as new concepts, as the underlying cause and source of inevitable irreversibility since absolute-0K temperature is unfeasible (3LT). Since *all real, irreversible processes generate heat and entropy* due to unavoidable dissipation of work and/or WP to heat (ultimately instigated by the “*thermal roughness*” as elaborated and named here), and *all ideal, reversible processes conserve entropy*, then, there is *no other process left* to miraculously generate WP without a due WP source. Furthermore, no “*imaginary process*” could destroy (or annihilate) entropy, since *it would be a “self-reversal of dissipation impossibility”* and *contradiction impossibility against the natural forcing*—it would imply *self-generation of non-equilibrium (and its WP)*; therefore, rendering a *logical proof of indestructibility of entropy* (the 2LT).

In the following Section 5, “*Carnot maximum efficiency, Reversible equivalency, and Work potential,*” Sadi Carnot’s ground-breaking contributions of reversible processes and heat-engine cycle maximum efficiency were put into historical and contemporary perspective. Furthermore, it has been argued that *Carnot’s contributions are among the most important developments in natural sciences*.

The proof by “*contradiction-impossibility*” of an established fact is, by definition, the logical proof of the stated fact. If a contradiction of a fact is possible then that fact would be void and impossible. It is illogical, absurd, and *impossible to have both, “the one-way and the opposite-way.”* For example, if heat self-transfers from higher to lower temperature, it would be “*contradiction-impossibility*” to self-transfer in the opposite direction. *All reversible processes (including cyclic processes) under the same conditions must have equal and maximum efficiency, as demonstrated by relevant “contradiction impossibility.”*

As a matter of fact, the reversible processes and cycles were a priori “specified” as ideal, with maximum possible efficiency, with a priory 100% “*2LT reversible-efficiency,”* not dependent on their design or mode of operation (independent on their quasi-stationary cyclic path or any other, reversible stationary process path). Actually, as the ideal “*work-extraction measuring-devices*”, all reversible processes and cycles determine, not their efficiency *per se*, but in fact, they determine the WP (as % or ratio efficiency with reference to relevant total input-energy) of an energy-source system with another reference system (such as the two thermal reservoirs with the Carnot cycle, so their WP ratio is dependent on their temperatures only). 

The *Carnot’s Equality* (*CtEq*), *Q/T* = *constant*, the well-known correlation, the precursor for the famous *Clausius Equality (CsEq)*, *CI*(*dQ/T*) = 0 (the cyclic integral for variable temperature reversible cycles), was specifically named here “as such” by this author in Sadi Carnot’s honor and to resample the *Clausius Equality (CsEq)* name.

In the succeeding Section 6, named here “Thermal Transformers: *Carnot-Clausius Heat-Work Reversible Equivalency (CCHWRE) concept”,* a notion of “true” *heat–work interchangeability* has been enlightened and named here, as an essential consequence of thermodynamic *reversible equivalency*.

The correlations, *Q_H_* ≡ *W_C_* + *Q_L_* and *Q_H_*/*T_H_* = *Q_L_*/*T_L_* = *W_C_*/(*T_H_* − *T_L_*), Equations (9) and (10), are much more important than they appear at first, since they represent the “*heat-work reversible equivalency and interchangeability*” in general, for all reversible steady-state processes not only for cycles (see Figure 7). Namely, heat *Q_H_* at high temperature *T_H_* is equivalent with the sum of heat *Q_L_* at lower temperature *T_L_* and Carnot’s work *W_C_*.

The energy of *thermal motion* (ThM) of ideal gas (IG) particles, *E_ThM_ = **E_th_** = N(k_B_T) = nR_u_T*, along with temperature also exhibited the pressure on any hypothetical or real boundary surface and, therefore, its energy may also be represented as mechanical (pressure) energy: *E_ThM_ = **E_me_** = PV*. Therefore, we may express the *IG equation of state* (i.e., the constitutive correlation of its mechanical and thermal properties) as the equivalence (“**≡**”) of the two forms of the same energy, *PV* ≡ *nR_u_T*, thus rendering its logical proof.

*Thermal transformers* were named and discussed by this author in 2004 [12], revisited later [17], and reiterated here in Section 6. They could be functioning as ideal, “*reversible thermal-transformers*.” Namely, the reversible heat transfer from higher *T_H_* to lower *T_L_* temperature with *W*, Carnot cycle work output; or in reverse, the reversible heat transfer from lower *T_L_* to higher *T_H_* temperature with *W*, Carnot cycle work input. Likewise, the real *thermal transformers*, combined power-and-heat cycles, and refrigeration cycles (including heat-pump cycles) also transfer heat from any to any temperature level, except for reduced efficiency due to unavoidable dissipation of WP to generated heat and entropy (Equations (5) and (7)).

Lastly, in Section 7, “‘*No Hope*’ *for the Challengers of the Second Law of thermodynamics,*” this author’s compelling arguments were presented, that “entropy can be reduced (locally, when heat is transferred out of a locality), but it cannot be destroyed by any means on any space or time scale of interest. A *“Perpetual-Motion Watch”* was presented as a trivial, *deceptive example.* Furthermore, three *Primary deception-structures (PDS)*, of hypothetical violation of the 2LT were classified by this author: *First PDS* (or “*dynamic [quasi-] equilibrium”*), *Second-PDS* (or “*structural [quasi-] equilibrium*”), and new, *third PDS* (or “*persistent-currents quasi-equilibrium*”). Lastly, critical discussions on the two selected publications by avid Challengers of the 2LT, one recent publication challenging the 2LT [24], and another, self-claimed as a “landmark paper”, experimentally challenging the validity of the 2LT [25], were presented.

The Challengers misinterpret the fundamental laws, present elusive hypotheses, and perform incomplete and misleading, biased experiments, always short of straightforward confirmation of their 2LT violation claims. That is why all resolved *Challengers’ paradoxes* and misleading violations of the 2LT to date have been resolved in favor of the 2LT and never against. We are still to witness a single, still open *Second Law violation*, to be verified and utilized.

The violation of the 2LT should be *the last* and *not the first* hypothesis to justify an unsolved phenomenon. It appears that the *Challengers* are misusing the elusive ‘*Entropy Law’* (2LT): *“Whoever uses the term ‘entropy’ in a discussion always wins since no one knows what entropy really is, so in a debate one always has the advantage”* (As lamented by John von Neumann). It would be more probable to assume that such structures are infinitesimally irreversible (or near reversible) and very slowly approaching true equilibrium while negligibly exhausting their own WP, or even hypothesize that they may be driven by negligibly stealthy, yet-to-be-discovered “cold-fusion-like” energy within atomic nucleus, or some stealthy WP from within or from the surroundings.

This treatise is concluded with the following:

***Challenge Point.*** *“Entropy of an isolated, closed system (or universe) is always increasing”, is a necessary but not sufficient condition of the second law of thermodynamics.* Entropy cannot be destroyed (annihilated), locally or at a time, and “compensated” by generation elsewhere or later. It would be equivalent to allowing rivers to spontaneously flow uphill and compensate it by a more downhill flow elsewhere or later. Thermodynamic (macroscopic) entropy is generated everywhere and always, at any scale (where it could be defined) without exception, and it cannot be destroyed by any means at any scale. Impossibility of entropy reduction by destruction should not be confused with a local entropy decrease due to entropy outflow with heat [13,17,18,19,20,21,22].

***Key Point 41.*** The *Second Law of Thermodynamics can be challenged, but not violated*—Entropy can be decreased, but not destroyed at any space or time scale. […]. The self-forced tendency of displacing non-equilibrium useful energy towards equilibrium, with its irreversible dissipation to heat, generates entropy, the latter is conserved in ideal, reversible processes, and there is no way to self-create useful energy from within equilibrium alone, i.e., no way to destroy entropy.”—[http://2LT.mkostic.com (accessed 5 July 2023)].

Furthermore, the time and spatial integrals of micro-quantities must result in macro-quantities, for the conservation laws to be valid. Therefore, claiming violation of the 2LT on micro-scale or special processes is questionable and due to lack of full comprehension of the 2LT, or due to lack of proper “tooling” (conceptual, analytical, numerical, or experimental limitations), or sometimes may be due to a desire for unjustified attention.

In reality, all processes must be at least *infinitesimally irreversible*, including underlining processes at equilibrium, the latter being an ideal state. The underlying mass-energy structures and processes within the “finest micro scales” are more complex and undetected at our present state of tooling and mental comprehension. However, their integral manifestation at macroscopic level, are more realistically observable and reliable, thus being the ultimate “check-and-balance” of microscopic and quantum hypotheses.

The fundamental physical laws are independent from any system structure or scale, and they should take primacy over any special analysis based on approximations and limitations of modeling of a system, its properties, and processes; and especially if based on “thought experiments” [22]. After all, micro- and sub-micro simulations and experimental analyses are also based on the fundamental laws, and therefore, they cannot be used to negate those fundamental laws. As the fundamental laws of nature and thermodynamics are expanded from simple systems in physics and chemistry, to different space and time scales and to much more complex systems in biology, life, and intelligent processes, there are more challenges to be comprehended and understood [17,18,19,20,21,22,23].

A summary of the Second Law of Thermodynamics (2LT), that emphasizes the spontaneous forced-displacement of nonequilibrium useful-energy (i.e., work-potential, WP) of mutually interacting systems, towards mutual equilibrium, with unavoidable WP dissipation to heat, accompanied with irreversible entropy-generation, at any space and time scale (locally or globally, where the macro-properties could be defined), without exception, is presented on Figure 10.

## Figures and Tables

**Figure 1 entropy-25-01106-f001:**
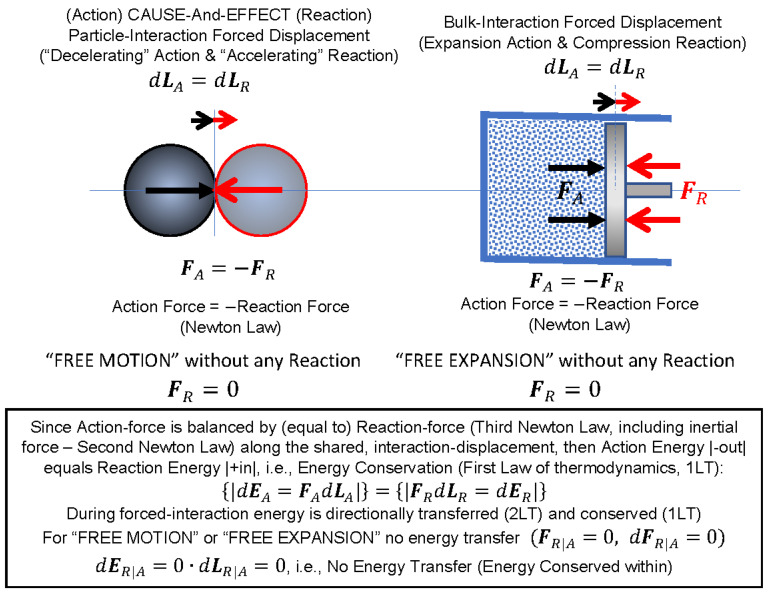
*Reasoning concepts of forcing and energy displacement*: Energy of a particle (or equivalent field particle) (**Left**) or bulk body (**Right**) will not change without forced interactions, i.e., *interactive forcing* (action–reaction) and *energy displacement* (energy transfer and conservation). A particle or bulk body in motion will uniformly move (**Left**) or freely expand (**Right**) unless interacting and exchanging energy with another particle or body. Elementary particle or ideal body interactions are reversible, but real, collective bulk-structure interactions of bounded collective-particles are irreversible due to dissipation of collective bulk, macro-energy within interacting micro-structures made up of interacting particles or equivalent field-particles.

**Figure 2 entropy-25-01106-f002:**
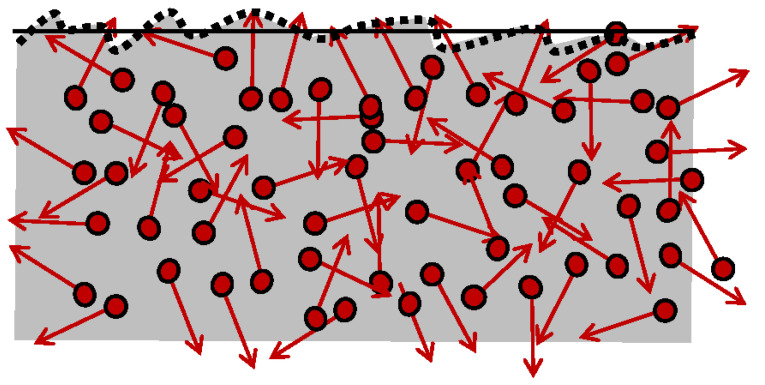
*Thermal roughness* and *thermal friction* are the underlying cause and source of unavoidable irreversibility (2LT) since absolute-0 K temperature is unfeasible (3LT), i.e., perpetual “smooth surface” is impossible. Real surface is always “*Dynamic-and-Rough*” (chaotic dotted-line) since it is impossible to have a “*Still-and-Smooth*” surface (plane solid-line) due to perpetual and unavoidable, dynamic “Thermal-Motion (ThM)” of “Thermal Particles (ThP)” always above unachievable, absolute-0 K temperature (3LT)).

**Figure 4 entropy-25-01106-f004:**
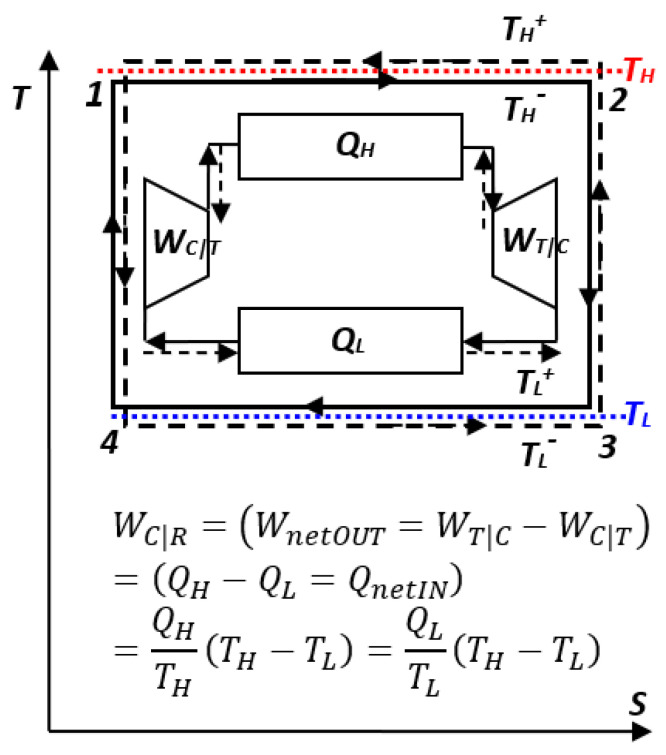
*Carnot* (*steam power*) *Cycle* (solid lines): heat *Q**_H_*** at *T**_H_*** is reversibly converted to work *W**_Max_** =W**_T_*** − *W**_C_*** and to *Q**_L_*** at *T**_L_***; and *Carnot reverse cycle* (dashed lines with reversed directions): work *W***_C*R*_**
*= W*_|***C***_ − *W*_|***T***_ and heat *Q**_L_*** at *T**_L_***, are converted to *Q**_H_*** at *T**_H_***. Thermal reservoirs’ high *T**_H_*** and low *T**_L_*** temperatures (dotted lines). *T* = turbine, *C* = compressor, *|X =* reverse of any *X*-quantity. All quantities are positive magnitudes [16,17].

**Figure 5 entropy-25-01106-f005:**
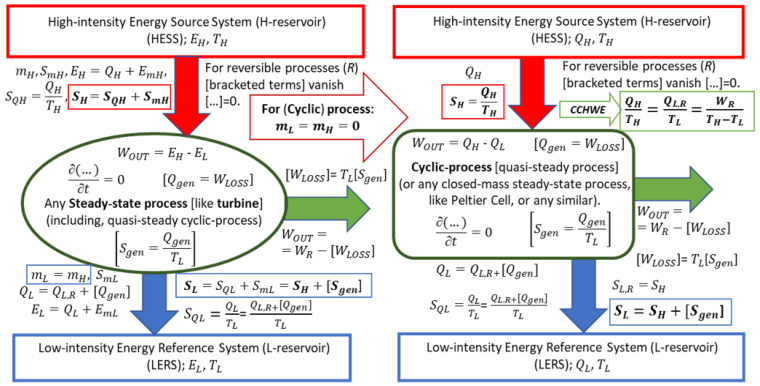
*Converting heat and internal energy to work*: In any *steady-state process* or *quasi-steady-cyclic process*, entropy input *S_H_*, with heat *Q_H_* (and with mass *m_H_* if any) at *T_H_* > *T_L_*, and if any irreversible generated entropy *S_gen_* within, must be discharged with heat *Q_L_* (and with mass *m_L_* if any), as entropy *S_L_* at *T_L_.* For ideal reversible process, $Q_{L,R}$ is “*not a loss but necessity*”, reducing maximum efficiency below 100%, such as in Carnot cycles (*Carnot Equality*). For real processes, irreversible work loss, $W_{LOSS}$ = $Q_{gen}$ = $T_{L}S_{gen}$, is due to dissipation of work to heat. For closed-mass and cyclic processes, *m_L_ = m_H_ =* 0, and for adiabatic turbine (*Q_H|L_ =* 0), *W_OUT_ = E_mH_ − E_mL_*.

**Figure 6 entropy-25-01106-f006:**
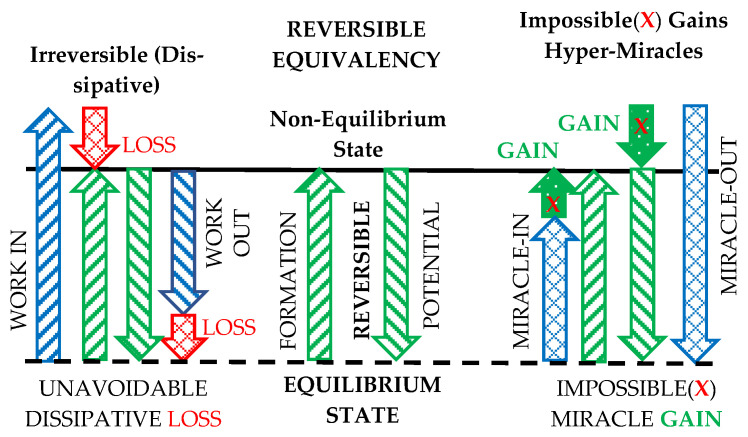
*Reversible equivalency:* Formation of “non-equilibrium state” requires “formation work-energy,” ideally all stored as “state work-potential (WP)” to be retrieved back in an ideal reversible process (Figure (**Center**), “*Reversible Equivalency*”: ideal formation work equal to work potential). Due to irreversible dissipation of work to heat (work loss), real formation work-in is bigger than stored WP, and retrieved useful work-out is smaller (**Left**). Formation of non-equilibrium state with less than its WP or obtaining more useful work than WP would require a “miracle Work-GAIN” without due WP source (violation of 2LT), being against natural forcing and existence of equilibrium, thus impossible (**Right**). Therefore, all reversible processes must be maximally and equally efficient [17,18].

**Figure 7 entropy-25-01106-f007:**
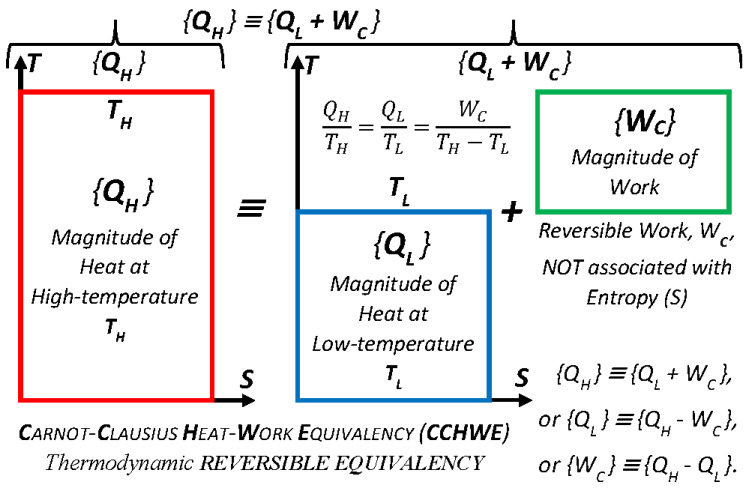
*Carnot–Clausius Heat–Work Reversible Equivalency (CCHWRE)* (as named here), established *interchangeability and related equivalency between “Heat-and-Work”*, based on the early work of *Carnot* (1824), that all reversible processes and cycles have equal and maximum efficiency, and among others, *Kelvin* and *Clausius’* meticulous work, around 1850s, that finalized the *thermodynamic temperature, reversible cycle efficiency, Carnot Equality, Clausius (In)Equality, Entropy,* and generalized the *Second Law of Thermodynamics (2LT)*.

**Figure 8 entropy-25-01106-f008:**
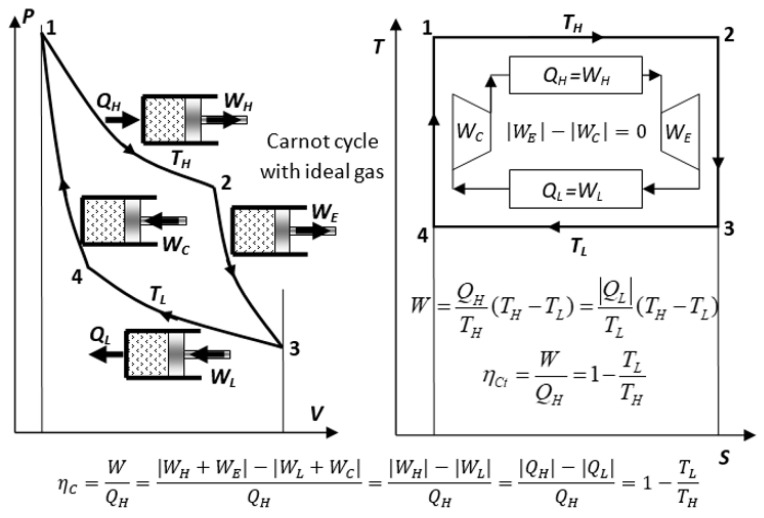
*Carnot cycle with ideal gas*: Isothermal expansion and compression’s works result in cycle net-work out, while adiabatic expansion and compression’s works cancel out, but they change temperatures required for reversible heat transfer.

**Figure 9 entropy-25-01106-f009:**
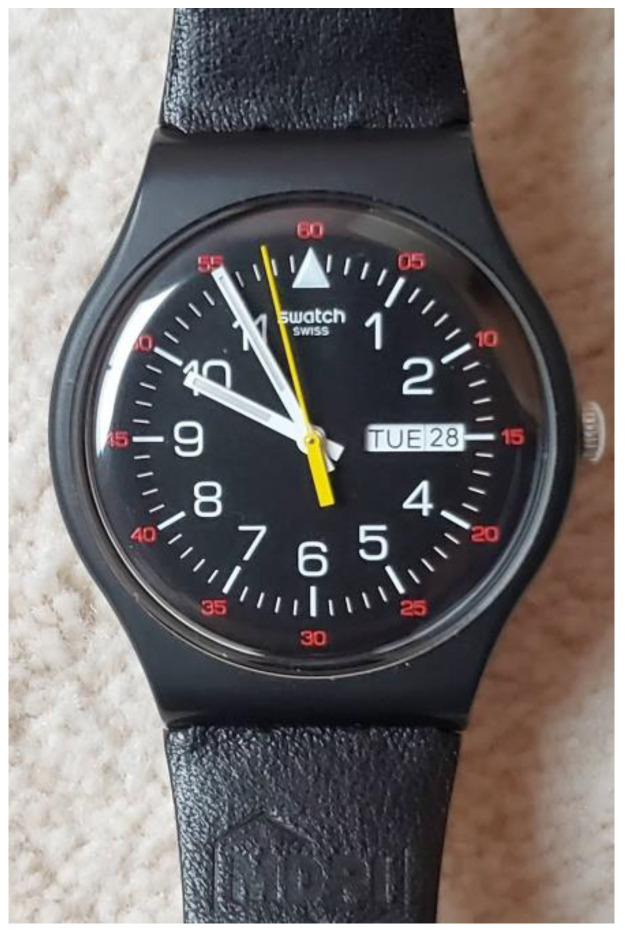
“*Perpetual-motion-like*” watch with 5–10 years battery life, as if its battery lasts forever. We could mistakenly hypothesize (as if we have proved experimentally), that it works without using energy (PMM1, 1LT violation), or it consumes energy from the surrounding thermal reservoir alone (PMM2, 2LT violation).

**Figure 10 entropy-25-01106-f010:**
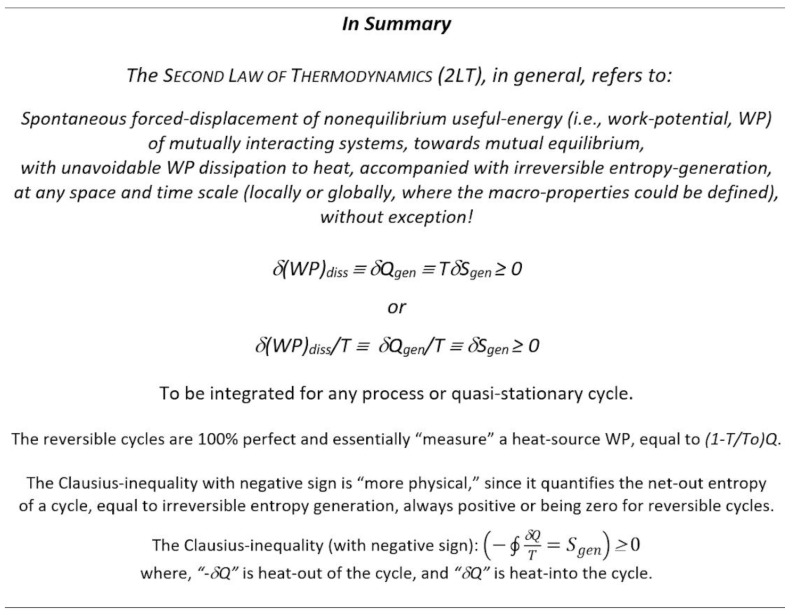
In summary—the Second Law of Thermodynamics (2LT).

## Data Availability

Not relevant for this study.

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
