# Peer review of "Reasoning and Logical Proofs of the Fundamental Laws: “No Hope” for the Challengers of the Second Law of Thermodynamics"

_entropy, 2023, doi:10.3390/e25071106_

Round 1

Reviewer 1 Report

In author's own words, the paper is a worthy presentation of "his lifelong endeavors and reflections with original reasoning and interpretations of the most critical and misleading issues in thermodynamics." The reviewer has only a few suggestions for the author to consider:

1. On p. 7 of Sect. 3, the word "mechanistic" is used in the sense of physics rather than of philosophy (as mechanistic philosophy). A statement to the effect may be in order.

2. In Sect. 5, False Point 1 makes a good point. It may be helpful to the readers by citing the excellent 1986 review article by Cropper: Cropper W H (1986). “Carnot’s function: Origins of the thermodynamic concept of temperature,” Am. J. Phys. 55 (2):120-129.

Incidentally, the opening sentence of Sect. 5 should be "It is the intention here to put..."

3. In Sect. 5, Key Point 19 makes an excellent point. It may be helpful to the readers when citing Clausius (1954_typo, it should be 1854) by mentioning Clausius (1854 Fourth Memoir).

4. The claim of "the maximum cycle efficiency depends on the temperature difference of the two reservoirs" being false of False Point 2 is in conflict with the statement of "Sadi Carnot reasoned that the maximum power efficiency has to be function of temperature difference..." on p.18. Please clarify.

5. The most important point of this review: There should be a False Point 4 with regards to the Available Energy Principle, the principle the theory of exergy is based on. The principle is not a law of nature as pointed out by Planck and Uffink: 

Planck commented, "The real meaning of the second law has frequently been looked for in a “dissipation of energy”… [But] there are irreversible processes in which the final and initial states show exactly the same form of energy … They occur only for the reason that they lead to an appreciable increase of the entropy" (Planck M (1969) Treatise on Thermodynamics, 3rd edition. (Dover, New York): 103–104).

Uffink, the historian of science, in a detailed analysis of the second law literature, commented on the issue spot-on, "Before Planck’s work there were also alternative views. We have seen that Kelvin attributed irreversibility to processes involving special forms of energy conversion. This view on irreversibility, which focuses on the ‘dissipation’ or ‘degradation’ of energy instead of an increase in entropy was still in use … Planck’s work extinguished these views …" (Uffink J (2001) “Bluff your way in the Second Law of Thermodynamics.” Stud Hist Phil Sci Part B: 2001 Studies in Hist and Philo of Modern Physics 32 (No. 2):42-43).

As per Planck and Uffink, the degradation of available energy, though spontaneous, is not universal. The assertion that it is a universal law is false. The author has the opportunity here to make this point if he chooses to. The reviewer would be glad to elaborate more on this issue. 

Author Response

See Attached file.

I like to thank all Reviewers for their constructive comments and suggestions that helped in further updates, clarifications with corrections, and hopefully improvement of the manuscript.

I have updated the manuscript with further clarifications to the questions raised by the three Reviewers, and with additional 5 References they suggested. Some of my “creative-ideas” are in “disagreements with established beliefs” of renowned authors (Planck) or respected authors (Cullen, Cropper), namely some “thermal-energy and entropy-generation concepts” (Internal energy “Pond analogy” etc.) … that I clarified and inferred to be misplaced/erroneous. These may be among the most valuable contributions to be further pursued. I have updated and structured the challenged-issues with more specifics, including new sub-sections, new Key-Points and new False-Points, with a hope to instigate future constructive criticism and resolution of the open issues.

Reviewer 2 Report

The author is congratulated for the very interesting and rich article. Some optional minor points to be addressed or better explained or reasoned in the text.

Page 2 “Thermodynamics, as the science of energy and entropy, is the most fundamental discipline, and as such, it encompasses all existence in space and transformations in time, in nature.”

Some might say quantum mechanics is the most fundamental discipline.

Page 3 “…and it is argued that the Carnot’s contributions are among the most important developments in natural sciences.” In this line of thought, it might be worth citing what Richard Feynman declares about Carnot in his Lectures.

Page 3 Please explain how energy density is different from absolute temperature.

Page 6 It might be a good idea to remove the previous to last line in Table I (the line with Etc. only).

Page 11 “However, this author does not agree with such and similar accounts [7] since the quality of such internal energies is not the same (i.e., different WPs and entropies; the 2LT).” Disagreement not clear, i.e., not clear how to distinguish the energy (property) transferred by heat (interaction) or work (interaction). See the analogy with water in a pond, in the book by Callen.

Page 19 “waisted” for heat transfer per se, since during the real… Please write wasted.

Page 24 “…it is evident that heat and work are interchangeable and truly (reversibly) equivalent as follows:” It may be worth it to cite the book by Gyftopoulos and Beretta, where it is claimed that a rigorous and clear distinction between heat and work is advanced, thus expressing a different point of view from the author.

Page 31 Section 7.3 seems unnecessary, or perhaps it could be shortened.

English writing is good, might want to read again and edit minor mistakes.

Author Response

See attached file.

I like to thank all Reviewers for their constructive comments and suggestions that helped in further updates, clarifications with corrections, and hopefully improvement of the manuscript.

I have updated the manuscript with further clarifications to the questions raised by the three Reviewers, and with additional 5 References they suggested. Some of my “creative-ideas” are in “disagreements with established beliefs” of renowned authors (Planck) or respected authors (Cullen, Cropper), namely some “thermal-energy and entropy-generation concepts” (Internal energy “Pond analogy” etc.) … that I clarified and inferred to be misplaced/erroneous. These may be among the most valuable contributions to be further pursued. I have updated and structured the challenged-issues with more specifics, including new sub-sections, new Key-Points and new False-Points, with a hope to instigate future constructive criticism and resolution of the open issues.

Reviewer 3 Report

Referee report Kostic:

“Reasoning and logical-proofs of the fundamental laws” submitted to ENTROPY in spring 2023

According to the author, the article is a comprehensive of “lifelong endeavors and reflections with original reasoning and reinterpretations of the most critical and misleading issues in thermodynamics”. A long list of novel abbreviations is introduced for known of new terms, a minimum of equations and a handful of plots for illustration of the ideas and examples are given. Historical remarks are made about the roles of Carnot, Kelvin and Clausius. A few new concepts are introduced like thermal roughness others.

A list of common energy forms and thermodynamic forces is shown in Table 1. In text books, they are used to formulate the Gibbs fundamental equation which connects each change of energy with the change of extensive quantities at given intensive quantities (forces). Here, however, only verbal arguments and statements are given to show their role in constituting processes and equilibria.

“Reasoning logical-proof of the fundamental laws,” is the title of section 3 where the treatment of particle collisions is used to prove (or at least illustrate) energy conservation (first law) due to Newton laws, which is correct, and directionality (second law), which makes no sense. The second law is about large systems only, even the treatment of collisions along the argumentation of Boltzmann’s H-theorem would be flawed, and the second law is known to hold for a much wider range of systems beyond simple colliding particles. The author seems to argue that the two laws become clear already in simple Newtonian dynamics, but this is not shown here.

In section 4, the dependency of a thermodynamic state on the way its energy is increased (heat or work) is trivial, so are the following reasonings about “indestructability of entropy” and “thermal roughness”. They would not help the reader to better understand thermodynamics.

Section 5 is dedicated to Carnot cycles and their efficiency, emphasizing the pioneering role of Carnot long before the later completion by Kelvin and Clausius.

In section 6, the heat-work equivalency which was the successful new concept of Clausius, is discussed (and partially ascribed to Carnot as an implicit idea). It is demonstrated on examples.

Section 7 of the manuscript examines challenges to the second law of thermodynamics, analyzing experimental evidence and critiquing claims that are deemed to be elusive or misinterpreted.

Overall, the manuscript provides a comprehensive analysis of the fundamental laws of thermodynamics, presenting original reasoning and reinterpretations of key concepts in defence against so-called challengers and any misinterpretations. The author's long-standing interest in the field is reflected in the extensive citations included in the manuscript, which was written as a “comprehensive treatise … written for the special occasion of the author’s 70th birthday”. While the manuscript does not present any new scientific discoveries, it serves as a valuable overview of the concepts of thermodynamics for non-specialists. However, the chosen format of strictly formulated key points and lengthy verbal explanations may not be the most engaging journalistic style for a general audience.

Author Response

(The authors gave the same response as above.)

Round 2

Reviewer 3 Report

The revised version 2 contains some more citations and additional text (keypoints) to explain in more detail the view on thermodynamic processes. In my first report I mentioned that parts of the article provide a new, interesting look on the history of thermodynamics which may be of interest for science-historical readers. For a general survey it may be acceptable that some statements refer well-known facts and reflections, some of which are perhaps trivial.

But I still believe that section 3 is a simplistic, non-scientific ‘derivation’ of the fundamental laws of thermodynamics from Newtonian mechanics. “Reasoning logical-proof of the fundamental laws” is the author’s notion for this kind of argumentation which in my opinion is rather confusing than helpful.

Overall, the MS is not very different from its previous shape. Its character is more science-historical than scientific, giving a comprehensive analysis with original reasoning and reinterpretations of key concepts in defense against so-called challengers and misinterpretations. The scientific level and the chosen format of strictly formulated key points and lengthy verbal explanations may not be the most engaging journalistic style for the readers of ENTROPY.